Corrected: Author correction

# NFAT primes the human *RORC* locus for RORγt expression in CD4$^+$ T cells

Hanane Yahia-Cherbal[1,2], Magda Rybczynska[1,6], Domenica Lovecchio [1], Tharshana Stephen[3], Chloé Lescale[4], Katarzyna Placek[1,7], Jérome Larghero[5], Lars Rogge [1] & Elisabetta Bianchi [1]*

T helper 17 (Th17) cells have crucial functions in mucosal immunity and the pathogenesis of several chronic inflammatory diseases. The lineage-specific transcription factor, RORγt, encoded by the *RORC* gene modulates Th17 polarization and function, as well as thymocyte development. Here we define several regulatory elements at the human *RORC* locus in thymocytes and peripheral CD4$^+$ T lymphocytes, with CRISPR/Cas9-guided deletion of these genomic segments supporting their role in RORγt expression. Mechanistically, T cell receptor stimulation induces cyclosporine A-sensitive histone modifications and P300/CBP acetylase recruitment at these elements in activated CD4$^+$ T cells. Meanwhile, NFAT proteins bind to these regulatory elements and activate RORγt transcription in cooperation with NF-kB. Our data thus demonstrate that NFAT specifically regulate RORγt expression by binding to the *RORC* locus and promoting its permissive conformation.

---

[1] Institut Pasteur, Immunoregulation Unit, Department of Immunology, Paris, France. [2] Université Paris Diderot, Sorbonne Paris Cité, Cellule Pasteur, Paris, France. [3] Institut Pasteur, Unité de Technologie et Service Cytométrie et Biomarqueurs (UTechS CB), Centre de recherche translationnelle (CRT), Paris, France. [4] Institut Pasteur, Genome Integrity, Immunity and Cancer Unit, Equipe Labellisée Ligue Contre le Cancer, Department of Immunology, Department of Genomes and Genetics, Paris, France. [5] Assistance Publique-Hopitaux de Paris, Hôpital Saint-Louis, Cell Therapy Unit and Cord Blood Bank; CIC de Biothérapies, CBT501 Paris, France. [6] Present address: Laboratoire Colloides et Matériaux Divisés, École supérieure de Physique et de Chimie industrielles, Paris, France. [7] Present address: Immunology and Metabolism, LIMES Institute, University of Bonn, Bonn, Germany. *email: elisabetta.bianchi@pasteur.fr

The ability of the immune system to mount efficient responses depends on the differentiation of naive CD4[+] T cells into functionally distinct T helper (Th) subsets, characterized by the secretion of specific "cytokine signatures". The original Th paradigm contrasted IFNγ-producing Th1 cells, important for host defense against intracellular pathogens, with Th2 cells, which produce IL-4, IL-5, IL-13 and are involved in the protection against parasitic infections. More recently, the range of effector CD4[+] T cells was expanded to include Th17 cells that secrete IL-17, IL-21, IL-22, and IL-26 in humans, and contribute to immune responses against extracellular bacteria and fungi[1].

Differentiation of Th subsets requires the integration of signals generated by the engagement of the T cell receptor (TCR) and by cytokines present at the time of stimulation: Th1 cells are generated in the presence of IL-12 or IFNγ, while IL-4 promotes Th2 differentiation[1]. The case of Th17 cells seems more complex: the minimal cytokine requirements for their generation in the mouse were described to be TGFβ and IL-6[2,3], with other pro-inflammatory cytokines such as IL-21 or IL-1β playing variable roles for Th17 generation both in the murine and the human system[4,5].

Differentiation is maintained by a network of subset-specific transcription factors and is stabilized through multiple cell divisions by epigenetic processes that regulate accessibility of regulatory chromatin regions, and promote heritable gene expression patterns. These epigenetic circuits can maintain cell identity when the initial signals driving differentiation are extinct[6]. Post-translational modifications of histone tails are key in influencing gene expression[7], and were shown to be important for T helper cell differentiation. The study of the epigenetic modifications of the *Ifng* locus in Th1 cells and *Il4* locus in Th2 cells showed that these genes are associated with permissive histone marks in the relevant lineage, while they are enriched with repressive modifications in the lineages that do not express the cytokine[8]. Similarly, in Th17 cells, the *Il17a* and *Il17f* loci are enriched for histone marks associated with a permissive chromatin conformation, such as Histone 3 acetylation (H3Ac) and Histone 3 Lysine 4 tri-methylation (H3K4me3)[9]. These histone modifications contribute to creating an open chromatin environment for the binding of transcription factors to these loci.

For each of these Th subsets, lineage-defining transcription factors, important for the establishment of the identity of the subset, have been described. Expression of T-bet in Th1, GATA3 in Th2 and RORγt in Th17 cells supports differentiation and function of the respective Th population[1]. Expression of these factors is not limited to the Th subset; in particular, RORγt was originally described as a thymus-specific isoform of the *RORC* locus, expressed selectively in double-positive (DP) thymocytes. *Rory*[−/−] mice showed reduced numbers of DP thymocytes and increased thymocyte apoptosis[10]. Mice that specifically lacked Rorγt expression also showed defective development of foetal lymphoid inducer cells (LTi), and lacked lymph nodes and Peyer's patches[11–13].

Downregulation of Rorγt expression is necessary for the maturation of DP thymocytes to the single-positive (SP) stage[14]. Rorγt expression remains suppressed in mature naive CD4[+] T cells, but can be re-expressed in the periphery in selected lymphocyte populations, including Th17 cells, which depend on Rorγt for IL-17 production[15]. The molecular mechanisms that regulate tissue-specific expression of RORγt are incompletely understood.

Major determinants of Th17 differentiation are the cytokines present at the time of priming. Of the several cytokines involved in the induction or maintenance of Th17 cells, IL-6, IL-21, and IL-23 all activate the transcription factor STAT3. Mutations in human *STAT3* in patients with hyper-IgE syndrome impairs

Th17 development[16,17]. Deletion of *Stat3* in mouse CD4[+] T cells results in the loss of IL-17 production and reduced levels of RORγt[5,18,19]. STAT3 may directly regulate RORγt transcription, as it binds to the first Rorγt intron in murine Th17 cells[19]. STAT3 also regulates RORγt indirectly, by inducing other transcription factors, such as HIF1 or the Soxt/Maf complex, which have been reported to bind and activate the murine Rorγt promoter[20,21].

STAT3-independent transcriptional pathways have been involved in RORγt induction: mice deficient for the NF-kB protein c-Rel showed compromised Th17 differentiation and reduced RORγt expression. Consistently, direct binding of NF-kB factors was detected at the murine *Rorc* locus and c-Rel and p65 were shown to directly activate the Rorγt promoter[22].

To date, the only transcription factors that have been implicated in thymic expression of *RORC* are E-proteins induced by pre-TCR signaling in late-stage DN (DN4) thymocytes[23]. Deletion of these factors reduced *RORC* expression in Th17 cells, indicating that E-box proteins may also stabilize *RORC* transcription in peripheral CD4[+] T cells[24]. Consistently, E-boxes in the RORγt promoter bound upstream stimulating factors USF1 and USF2 in the human Jurkat cell line[25].

These findings suggest that RORγt regulation is likely the result of molecular interactions within a multifactorial complex, whose exact components remain to be identified.

In this work we explore epigenetic and transcriptional mechanisms associated with human RORγt expression in thymocytes and in vitro differentiating Th17 cells, with particular attention for TCR-activated signaling pathways. We define genomic regions surrounding the RORγt promoter that undergo profound remodeling in thymocytes or in stimulated peripheral CD4[+] T cells. Our data demonstrate that the activation of NFAT family transcription factors plays an essential role in RORγt expression and promotes a permissive conformation at the RORγt promoter and upstream regulatory regions. These data support a model where non-specific TCR-mediated activation primes at Th lineage-specific loci an accessible chromatin conformation, which is further stabilized by subset-specific factors induced by polarizing cytokines, resulting in tissue-specific transcription.

## Results

**Remodeling of the *RORC* locus thymocyte development**. RORγt was first detected in murine double-positive thymocytes. RORγt and its isoform RORγ are encoded by the *RORC* locus, through the activation of alternative promoters, *RORC2* and *RORC*, respectively[22] (Fig. 1a). To analyze expression of RORγt in the human thymus, we sorted the following thymocyte populations showing progressive degrees of maturation: CD4[−]CD8[−]CD3[−] (double-negative, DN), CD4[+]CD8[−]CD3[−] (immature single positive, ISP), CD4[+]CD8[+] double positive with low CD3 expression (DP CD3low), CD4[+]CD8[+] double positive with higher CD3 expression (DP CD3int), and CD4[+] single positive or CD8[+] single positive cells (SP CD4 and SP CD8), which express the highest levels of CD3 (Supplementary Fig. 1a). *RORC* expression remained at background levels in all samples analyzed; *RORC2* expression started to increase at the ISP stage, peaked in DP cells, and dropped again in SP cells, remaining low in naive CD4[+] and CD8[+] T cells from peripheral blood (Fig. 1b).

We then asked whether selected *RORC2* expression in DP thymocytes entailed specific chromatin modifications at the *RORC* locus. Chromatin Immunoprecipitation (ChIP) assays coupled to RT-qPCR showed that the *RORC2* promoter undergoes permissive Histone 4 hyperacetylation (H4Ac) at the transition from the DN to ISP stage (Fig. 1c). H4Ac of the *RORC2* promoter persisted in SP CD4[+] cells that no longer express *RORC2*, as well as in naive CD4[+] cells from the

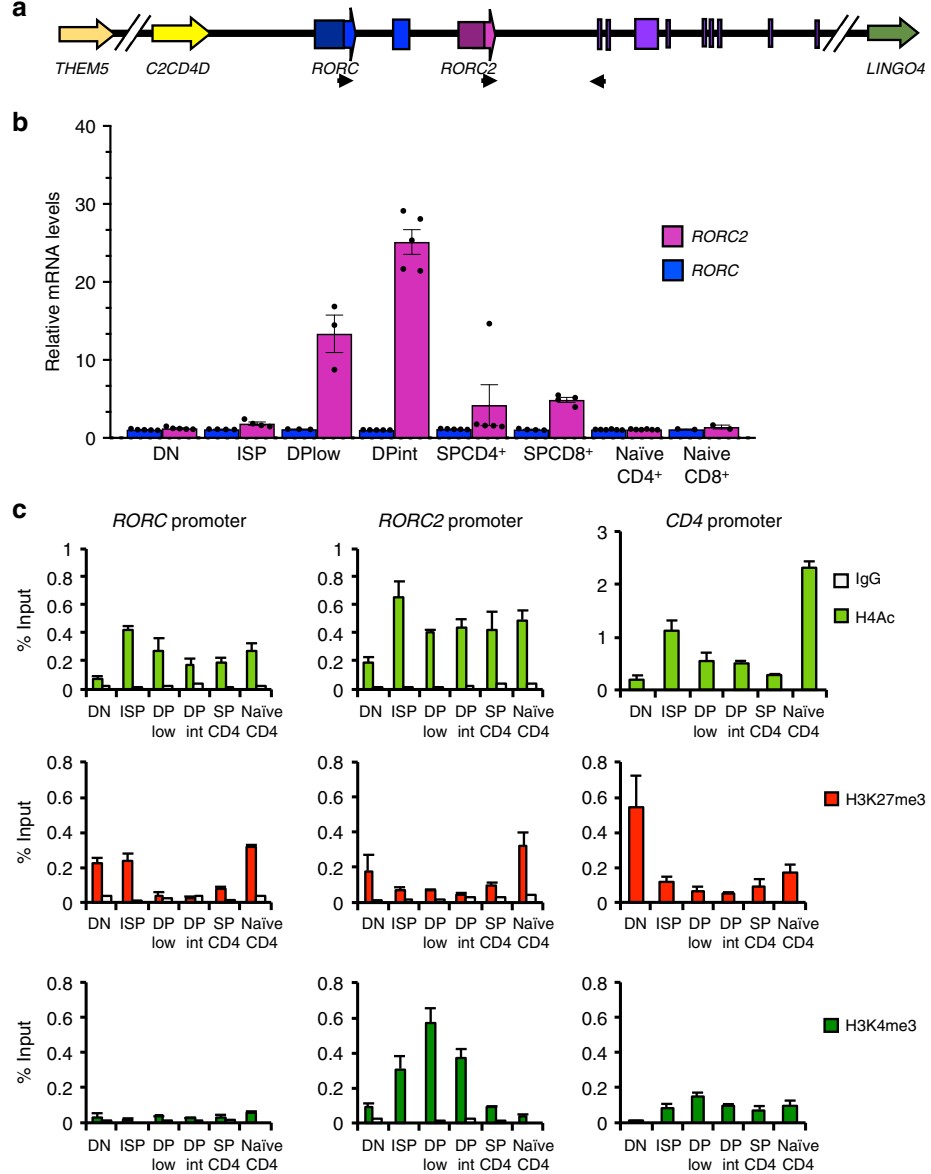

**Fig. 1** Remodeling of the *RORC* promoter during thymocyte development. **a** Scheme of the human *RORC* locus: transcription from the *RORC* promoter generates the RORγ isoform; the *RORC*2 promoter generates the RORγt isoform. Blue boxes: unique *RORC* exons; pink box: unique *RORC*2 exon; purple boxes: common exons. Arrows indicate the primers used for RT-qPCR amplification of the two isoforms. **b** *RORC* and *RORC*2 expression in sorted thymocyte populations, and naïve CD4$^+$ or CD8$^+$ T cells isolated from cord blood was quantified by microarray. Shown is average and standard deviation of five independent experiments. **c** Epigenetic modifications at the *RORC*, *RORC*2 and *CD4* promoters. ChIP was performed with antibodies against histone 4 acetylation (H4Ac, top); histone 3 lysine 27 trimethylation (H3K27me3, middle) and histone 3 lysine 4 trimethylation (H3K4me3, bottom), on sorted thymocyte populations, and in naïve CD4$^+$ T cells from cord blood, followed by RT-qPCR of the *RORC* promoters and the *CD4* promoter (as a quality control). ChIP with an irrelevant IgG antibody tested the specificity of binding (white bars). Data are expressed as percent of input and represent average and SD of three replicates. Source data are provided as a Source Data file

periphery. Increased acetylation at the ISP stage was paralleled by a decrease in the repressive Histone 3 Lys27 tri-methylation mark (H3K27me3), which was restored in peripheral naïve CD4$^+$ T cells, following the extinction of *RORC*2 expression, consistent with a role in consolidating suppression of *RORC*2 in these cells. The same pattern for these modifications was detected at the *RORC* promoter, despite lack of *RORC* expression, indicating that while these modifications may be important to increase general accessibility at the *RORC* locus, they do not directly reflect active transcription at this locus.

A more specific enrichment pattern was observed for tri-methylation of Histone 3 Lys4 (H3K4me3), a modification

associated with actively transcribed promoters[26]. The enrichment of this mark at the *RORC*2 promoter closely reflected *RORC*2 expression in the different thymocyte populations, with no increase in H3K4me3 at the *RORC* promoter at any stage of thymocyte development (Fig. 1b, c).

The discrete distribution of H3K4me3 was more evident when we extended the analysis to other conserved non-coding (CNS) regions (Fig. 2a, b): H3K4me3 was increased exclusively around the *RORC*2 promoter and at the first *RORC*2 intron, and only in cell populations where *RORC*2 was actively transcribed. This was in contrast to the changes in H4Ac and H3K27me3, which showed the same pattern of enrichment at all analyzed regions (Fig. 2b).

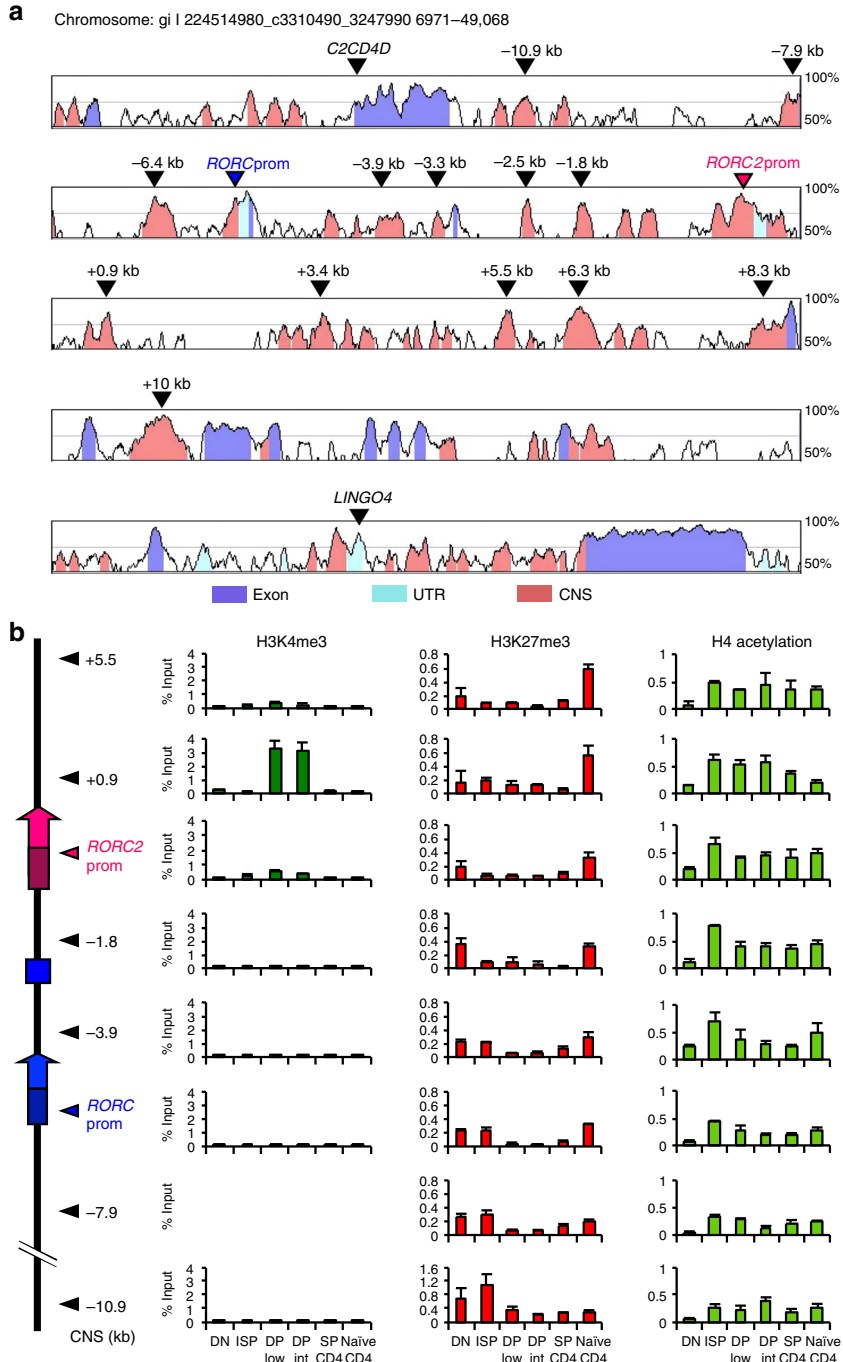

**Fig. 2** Global remodeling of the *RORC* locus during thymocyte differentiation. **a** Alignment of the human and murine *RORC* loci, obtained with the VISTA browser. The *Y*-axis displays the percentage of identity between sequences. Regions with >70% identity are marked in color: Conserved Non-coding Sequences (CNS) in salmon, 5′ and 3′ untranslated regions (UTR) in light blue and exons in purple. The arrowheads show the localization of the CNS analyzed in the ChIP assays. CNSs are labeled according to their distance from the *RORC2* Transcription Start Site. **b** The diagram on the left shows the organization of the *RORC* locus. The genomic regions amplified are marked with arrowheads. Enrichment of H3K4me3 (left panels), H3K27me3 (middle panels), and H4Ac (right panels) was analyzed by ChIP assay in DN, ISP, DP CD3low, DP CD3 intermediate, SP CD4 thymocytes, and naive CD4+ T cells from cord blood. Average and SD of three replicates are represented. Source data are provided as a Source Data file

These data show that the whole *RORC* locus undergoes extensive remodeling in thymocytes at the DN to DP transition, acquiring increased accessibility that allows local H3K4me3 and recruitment of the transcriptional machinery at the *RORC2* promoter. The enrichment in H4Ac is maintained after the extinction of *RORC*2 expression, suggesting that a more permissive conformation persists in naive cells, and may favor renewed *RORC*2 expression upon appropriate stimulation in peripheral T cells.

To further characterize putative regulatory sequences at the *RORC* locus, we performed a ChIP analysis for histone modifications associated with enhancer regions (Fig. 3). H3 Lys4 monomethylation (H3K4me1) is widely distributed and has been reported to mark enhancers independently of their activation[27]. We found that the H3K4me1 specifically increased at the DP stage and showed a diffused enrichment along the locus (Fig. 3, left column). The enrichment of H3 Lys27 acetylation (H3K27Ac)

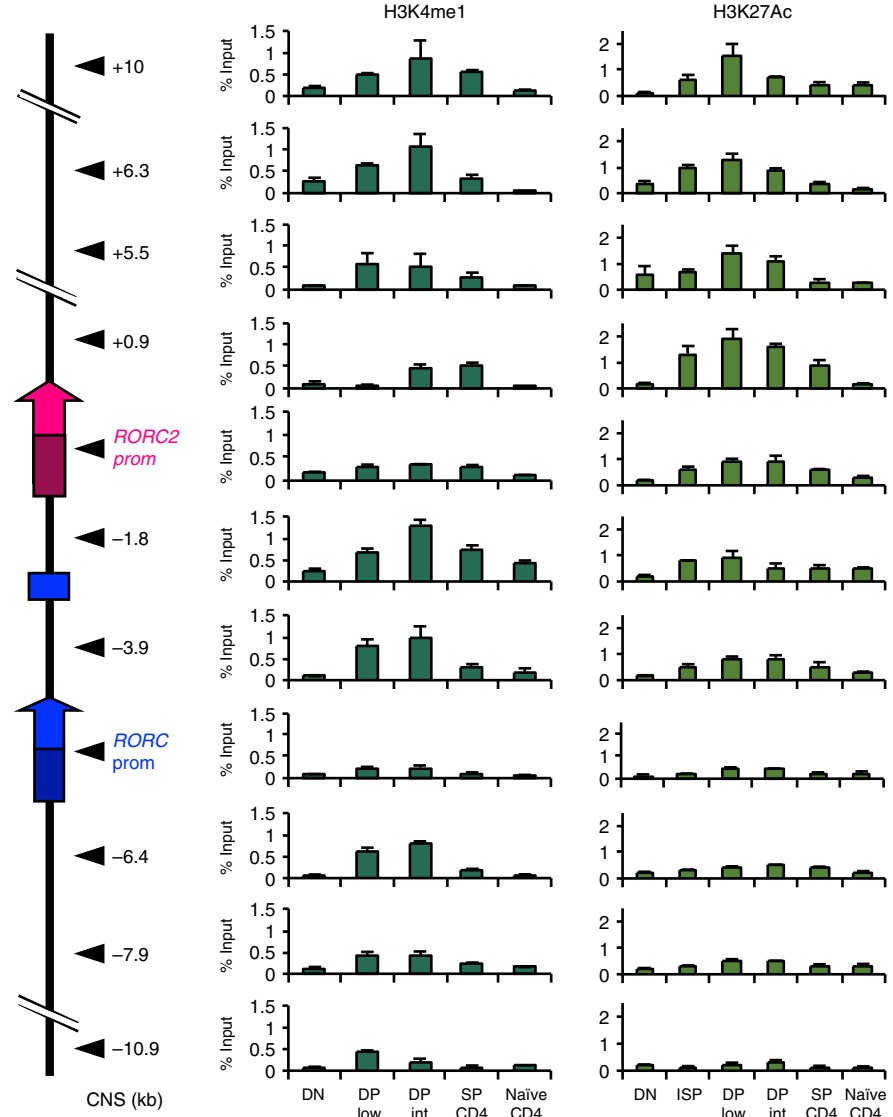

**Fig. 3** Enrichment of enhancer marks at the *RORC* locus in thymocytes. Enrichment of H3K4me1 (left panels) and H3K27Ac (right panels) was analyzed in different thymocyte populations, and in naive CD4$^+$ T cells. Data represent average and SD of three replicates. The diagram on the left shows the regions analyzed, defined by their distance from *RORC2* TSS. Source data are provided as a Source Data file

marks more selectively active enhancers, with a high degree of cell and tissue specificity[27]. At the *RORC* locus, H3K27Ac was enriched in ISP and DP cells at CNS located between the *RORC* and the *RORC2* promoter (CNS-1.8 kb and CNS-3.9 kb), as well as at several CNS within the *RORC2* introns (CNS+0.9 kb, +5.5 kb, +6.3 kb, +10 kb), suggesting that these regions may have regulatory function for *RORC2* expression. No H3K27Ac enrichment was observed upstream of the *RORC* promoter. Reflecting the decline in *RORC2* expression, this mark was progressively lost in SP and naive CD4$^+$ cells (Fig. 3, right column).

**Induction of *RORC2* in CD4$^+$ T cells is cyclosporine-sensitive**. In peripheral CD4$^+$ T cells, *RORC2* is re-expressed upon differentiation of naive cells to Th17 cells. We asked whether *RORC2* expression in peripheral CD4$^+$ T cells involves the same genomic regions and epigenetic processes observed during thymic expression of *RORC2*. To study the induction of *RORC2* during the early stages of human Th17 differentiation we used an in vitro system that allows us to dissect the signaling pathways involved in *RORC2* expression. Cord blood-derived naive CD4$^+$ T cells were

stimulated through the TCR and the CD28 receptor, in the presence of different combinations of cytokines. We analyzed the expression of *RORC* and *RORC2* by Real-time quantitative PCR. While optimal induction of *RORC2* expression was obtained in the presence of TGFβ, IL-1β, IL-21, and IL-23 ("Th17" inducing conditions[28]), stimulation through the TCR alone was sufficient to induce low levels of *RORC2* mRNA, which were further increased by the addition of TGFβ (Fig. 4a). Induction of *RORC2* was already visible 4 h after stimulation (Supplementary Fig. 1b). TCR stimulation in the presence of TGFβ or Th17-inducing cytokines, also induced *RORC* transcription, although the mRNA levels for *RORC* in CD4$^+$ lymphocytes were at least one hundred-fold lower than those detected in human hepatocytes (Supplementary Fig. 1c). TGFβ alone had no effect on *RORC2* expression, underscoring the central role of TCR stimulation (Fig. 4a).

A prominent signaling pathway activated downstream of TCR engagement is the calcineurin/NFAT pathway. TCR stimulation activates the phosphatase calcineurin, which dephosphorylates the regulatory domain of transcription factors of the NFAT family, promoting their nuclear translocation, binding to DNA and transcriptional activation[29,30]. Stimulation of cord blood

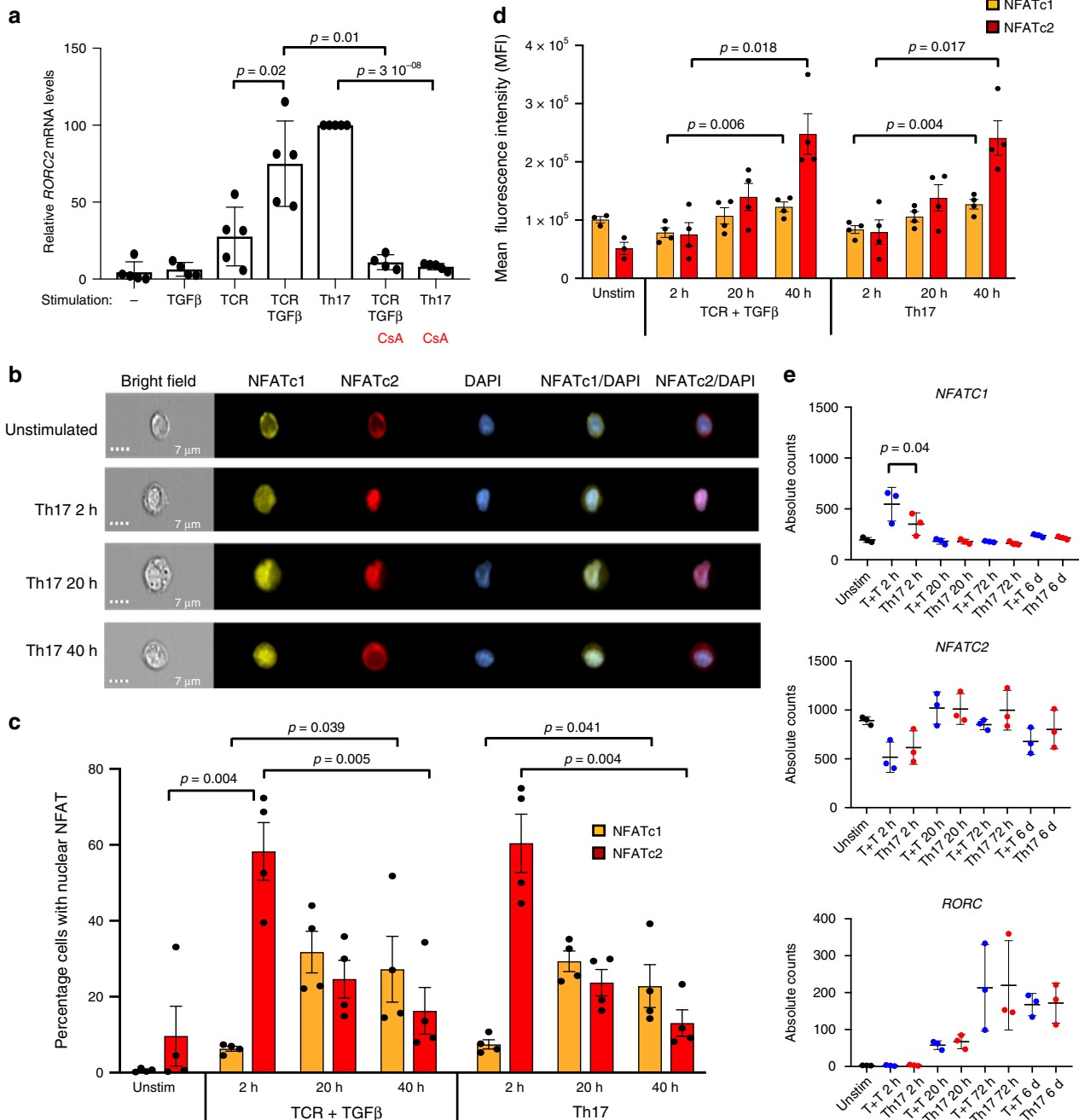

**Fig. 4** NFAT/calcineurin regulates *RORC*2 expression in peripheral CD4[+] T cells. **a** *RORC*2 expression is calcineurin-dependent. Cord blood CD4[+] T lymphocytes were stimulated for 20 h. *RORC*2 expression was quantified by RT-qPCR and represented as fold change relative to Th17 conditions. Average and SD of 5 experiments. **b** Cord blood CD4[+] T cells were stimulated in Th17-inducing conditions for the indicated times, and the distribution of NFAT proteins was analyzed by Imaging Flow Cytometry. Representative images are shown for staining for NFATc1, NFATc2, DAPI, and the overlap of NFATc1 or NFATc2 with DAPI. Scale bar: 7 μm. **c** The percentage of cells with nuclear NFAT translocation was calculated using a Similarity Score, and the average of 4 independent experiments was plotted for cells stimulated as shown. Bars indicate SEM. **d** The average mean fluorescence intensity of the NFAT positive cells was plotted for the same samples analyzed in **c**. **e** Time-course analysis of absolute mRNA counts of *NFAT* and *RORC* measured by nCounter technology in stimulated cord blood CD4[+] T cells. Shown are average and SEM from 4 independent experiments. *p*-values resulted from paired two-tailed *t*-test. "−": unstimulated, CsA: Cyclosporine A, Unstim: unstimulated sample, T + T: TCR + TGFβ, 6d: 6 days. Source data are provided as a Source Data file, raw image files are available directly from the authors

CD4[+] T cells in the presence of the calcineurin inhibitor, Cyclosporine A (CsA), resulted in significant inhibition of *RORC*2 expression, indicating the importance of the calcineurin/NFAT pathway in *RORC*2 induction (Fig. 4a).

NFAT family proteins share a conserved DNA-binding REL-homology domain and recognize similar binding sites, but can also show different preferences for specific DNA sequences, resulting in selective transcriptional activation[30]. NFATc1 (NFATc or NFAT2), NFATc2 (NFATp or NFAT1), NFATc3 (NFAT4), and NFATc4 (NFAT3) require calcineurin for their activation, while NFAT5 is activated by osmotic stress and is calcineurin-independent. In addition, several isoforms with

different N- and C-termini are generated by alternative promoter usage and splicing. We focused our analysis on NFATc1 and NFATc2, since these calcineurin-dependent factors were reported to be expressed in peripheral T lymphocytes[29].

To explore the role of the NFAT pathway in the early phases of human Th17 differentiation, we studied the dynamics of NFAT nuclear translocation in differentiating cord blood CD4[+] T cells. Imaging Flow Cytometry analysis demonstrated a strong nuclear accumulation of NFATc2 in cells stimulated in Th17-differentiating conditions, or through the TCR in the presence of TGFβ (Fig. 4b, c, Supplementary Fig. 2), at an early time point, preceding the increase in RORC2 mRNA levels (Supplementary Fig. 1b). Nuclear translocation of NFATc1 followed a slower kinetics, and was less pronounced than NFATc2 translocation. Both Th17 and TCR + TGFβ stimulation resulted in an increase of NFATc2 mRNA and protein (Fig. 4d, Supplementary Fig. 1d, e), which persisted for the time of observation, while the transient increase of NFATc1 mRNA (Fig. 4e) was accompanied by a smaller increase in NFATc1 mean fluorescence intensity.

**NFAT binds to the RORC locus in stimulated CD4[+] T cells.** Given the involvement of calcineurin activation in RORC2 expression, and the strong early nuclear accumulation of NFATc2 in stimulated cord blood cells, we asked whether NFAT factors directly regulate RORC2 transcription by binding to the human RORC locus. Binding of NFATc2 to the murine Rorc gene was described by Kim and coworkers to occur in Th17-polarizing conditions[31]. We tested binding of NFAT proteins to the RORC locus in vivo in cord blood CD4[+] T cells using ChIP assays, in unstimulated conditions, and in conditions that induce RORC2 expression. ChIP assays performed with anti-NFATc1 antibodies showed that in stimulated cells NFATc1 binds to the RORC2 promoter and to the putative regulatory regions upstream of the RORC2 promoter, CNS-1.8 kb and CNS-2.5 kb (Fig. 5a). NFATc1 binding could be already detected in cells stimulated through the TCR (Supplementary Fig. 3a), it increased in the presence of TGFβ, and was cyclosporine-sensitive. However, NFATc1 binding was strongly reduced in cells stimulated in Th17-inducing conditions (Fig. 5a and Supplementary Fig. 3a).

To the contrary, enrichment of NFATc2 binding to the RORC locus could be detected at these same regions both in cells stimulated through the TCR in the presence of TGFβ, and in Th17 inducing conditions (Fig. 5b). Binding was inhibited by CsA treatment (Fig. 5b). Only limited NFAT binding was detected at the RORC promoter (Fig. 5a and Supplementary Fig. 3b). NFAT did not bind at CNSs in the first RORC2 intron (CNS+0.9 to CNS+8.3), which also display marks of active enhancers (Figs. 3 and 6b).

In silico analysis of the DNA sequence at the NFAT-bound regions revealed the presence of several NFAT consensus binding sites (Fig. 5c and Supplementary Table 3). We used an in vitro DNA affinity assay to test whether these sequences bind NFAT proteins. We incubated extracts from the Jurkat E6.1 T cell line with biotinylated oligonucleotides corresponding to the predicted NFAT binding sites, and analyzed proteins bound to these oligos by SDS-PAGE and Western blotting. As a positive control, we used a characterized NFAT-binding sequence from the IL-2 promoter[32], and as a negative control a sequence from the RORC locus that lacks any predicted NFAT binding sites (Supplementary Table 3). We found that both NFATc2 and NFATc1 bind strongly to all four predicted NFAT sites in the RORC2 promoter (Fig. 5c, oligos 4 to 7, Supplementary Fig. 3c). We also observed binding to oligos 1 and 2, which correspond to the upstream NFAT sites in CNS-1.8 kb and CNS-2.5 kb. No binding was detected to oligo 3, which contains a predicted NFAT site located in a non-conserved region upstream of

the RORC2 promoter (Fig. 5c, Supplementary Fig. 3c). Specificity of binding was supported by the fact that preincubation with an unbiotinylated oligo containing a wild type NFAT binding site (W) abolished binding of NFAT proteins to the RORC oligos, while addition of an oligo containing a mutated NFAT binding site (M) did not compete with the RORC2 oligos for binding to NFATc1 or NFATc2 (Fig. 5d, Supplementary Fig. 3d and Supplementary Table 3).

A similar pattern of binding to the RORC oligos was detected when protein lysates were obtained from primary CD4[+] T cells isolated from cord blood (Fig. 5e, Supplementary Fig. 3e). These findings confirm that NFAT proteins can bind directly to several consensus sequences at transcriptional regulatory regions of the RORC locus.

**NFATc2 promotes transcription from the RORC2 promoter.** To test the functional outcome of NFAT binding to the RORC locus, we generated firefly luciferase reporter vectors containing either the RORC2 promoter alone, or the RORC2 promoter together with the CNS-2.5 kb and CNS-1.8 kb putative regulatory sequences, and we tested their activity in reporter gene assays. The RORC2 promoter construct transfected in HEK293T cells or in Jurkat cells showed constitutive transcriptional activity (Fig. 5f), similarly to what was reported for the murine RORγt promoter[22]. This activity was abolished by inversion of the promoter sequence (Fig. 5f) and was not inhibited by CsA treatment, indicating that it is independent of NFAT proteins (Supplementary Fig. 3f). To test the role of NFAT in the transcriptional activation of the RORC promoter, we chose the HEK293T cell line, which does not express endogenous NFATc1 nor NFATc2 proteins (Supplementary Fig. 3i). Co-expression of NFATc2 significantly increased RORC2 promoter activity, however no significant effect on RORC2 promoter activity was detectable upon overexpression of the inducible NFATc1A protein, or of the longer isoform NFATc1C, which, like NFATc2, contains an additional C-terminal transactivation domain (Fig. 5g). We then asked whether binding of NFAT to the putative regulatory sites upstream of the RORC2 promoter could further increase promoter activity. The fold increase in transcriptional activity obtained in HEK293T cells by co-transfection of NFATc1 or NFATc2 with the reporter construct containing CNS-2.5 kb and CNS-1.8 kb was not significantly different from the fold-increase obtained with the construct containing the RORC2 promoter alone (Supplementary Fig. 3g). These findings suggest that NFAT binding to the upstream sites does not directly contribute to enhancing transcription from the nucleosome-free RORC2 promoter, raising the possibility that it may play a different role in the context of chromatin.

**The RORC locus undergoes calcineurin-dependent remodeling.** To explore additional mechanisms of action of NFAT proteins in RORC2 induction, we asked whether NFAT recruitment could be involved in establishing a transcriptionally active conformation of the RORC locus. NFAT has been shown to induce permissive chromatin modifications by recruiting histone modifying enzymes, such as the histone acetyltransferases p300/CBP, to target genes[33]. To investigate whether the NFAT pathway plays a role in chromatin remodeling at the RORC locus we analyzed the effect of calcineurin inhibition on histone modifications at the RORC and RORC2 promoters, in unstimulated and stimulated CD4[+] T cells isolated from cord blood (Fig. 6a). TCR stimulation alone was sufficient to induce an enrichment of H3K4me3, and hyper-acetylation at the RORC2 promoter (Fig. 6a). Both H3K4me3 and H4Ac levels were further increased by the addition of TGFβ or of

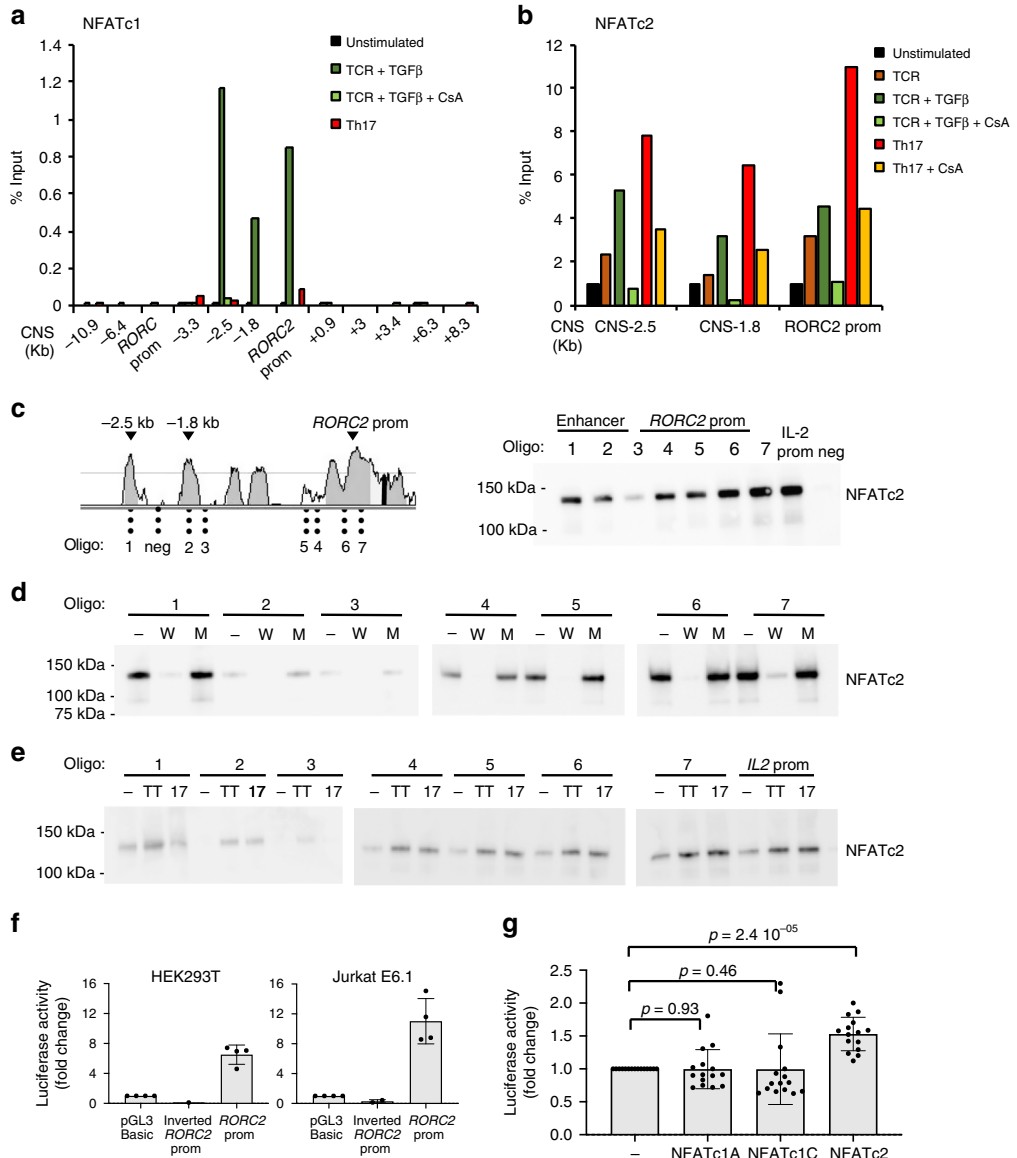

**Fig. 5** NFAT binds to the *RORC* locus in stimulated CD4$^+$ T cells. **a** NFATc1 binds to the *RORC* locus in stimulated CD4$^+$ T cells. ChIP was performed on cord blood CD4$^+$ T lymphocytes, followed by RT-qPCR of the indicated genomic regions. **b** NFATc2 binds to the *RORC* locus in Th17 cells. ChIP was performed as in **a**. Shown are one representative experiment of 3, average of three technical replicates. **c** Left: human/murine alignment of the *RORC*2 promoter and upstream conserved region (CNS-2.5 kb and -1.8 kb), with the position of the predicted NFAT binding sites tested in the DNA affinity capture assay. Right: DNA affinity capture assay on lysates from Jurkat E6.1 cells stimulated with PMA/ionomycin for 30 min. The western blot of DNA-bound proteins was probed with anti-NFATc2 antibody. *IL-2 prom*: characterized NFAT site in the *IL-2* promoter. *neg*: sequence in the *RORC* locus with no predicted NFAT binding sites. **d** DNA affinity competition assay. Jurkat E6.1 lysates were incubated with biotinylated NFAT binding oligos, together with non-biotinylated oligos containing a wild-type (W) or mutated (M) NFAT binding site. **e** DNA affinity capture assay on cord blood CD4$^+$ T cells stimulated in the indicated conditions ("−": unstimulated, TT: TCR + TGFβ; 17: Th17 polarizing condition). **f** Activity of the *RORC*2 promoter in cell lines. HEK 293 T and Jurkat E6.1 cells were transfected with the indicated reporter constructs. Normalized luciferase activity is represented as fold change over the activity of pGL3 basic. Average and SD of 4 experiments. **g** NFATc2 activates transcription from the *RORC*2 promoter. HEK293T cells were transfected with the *RORC*2 promoter reporter and the shown NFAT constructs. Luciferase activity is represented as fold change relative to activity of the *RORC*2 promoter alone (first bar). Average and SD from 15 transfections. Two-tailed t-test was performed as shown. Source data are provided as a Source Data file

Th17-inducing cytokines. Contrary to what observed in the thymus, H4 hyperacetylation in stimulated naive CD4$^+$ T cells was restricted to regions surrounding the *RORC*2 promoter, and not detectable at the *RORC* promoter (Fig. 6b). The enrichment of these chromatin modifications was impaired by CsA, suggesting that activation of the calcineurin pathway is important for the permissive remodeling of the *RORC* locus. TCR activation alone also induced CsA-sensitive enrichment of H3K27 acetylation at the *RORC*2 promoter, as well as the surrounding CNSs (Fig. 6a, b).

To test whether p300 and/or CBP may be involved in the calcineurin-dependent acetylation process, we analyzed the binding of these proteins to the *RORC* locus in ChIP assays. Both p300 and CBP were found to bind to the *RORC*2 promoter in stimulated, but not resting, T cells, and the binding was decreased by treatment with CsA, suggesting that NFAT activation promotes recruitment of these acetyltransferases to the *RORC* locus (Fig. 7a, Supplementary Fig. 3j, k). Calcineurin-dependent binding of p300 was also observed at CNS-2.5 and

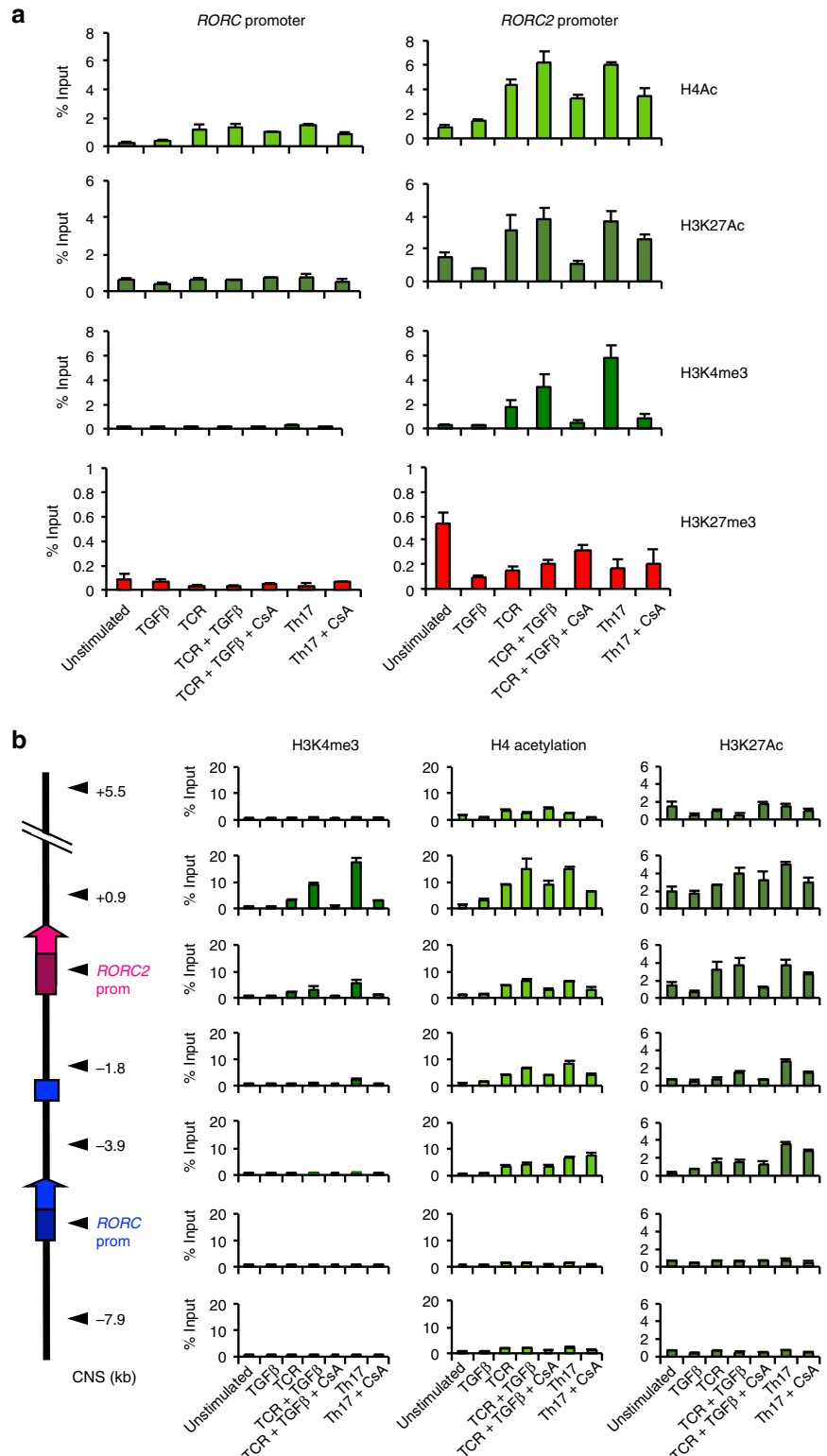

**Fig. 6** NFAT activation promotes epigenetic remodeling at the *RORC2* locus. **a** Epigenetic modifications at the *RORC2* promoter in cord blood CD4[+] T lymphocytes are calcineurin-sensitive. Cord blood CD4[+] T lymphocytes were stimulated 20 h as indicated. ChIP was performed with antibodies against H4Ac, H3K27Ac, H3K4me3 and K3K27me3. **b** ChIP was performed with antibodies recognizing the following modifications: H3K4me3, H4Ac and K3K27Ac. The diagram on the left shows the regions analyzed. Enrichment is represented as percent of input DNA. CsA: Cyclosporine A. Data represent average and standard deviation of three replicates. Source data are provided as a Source Data file

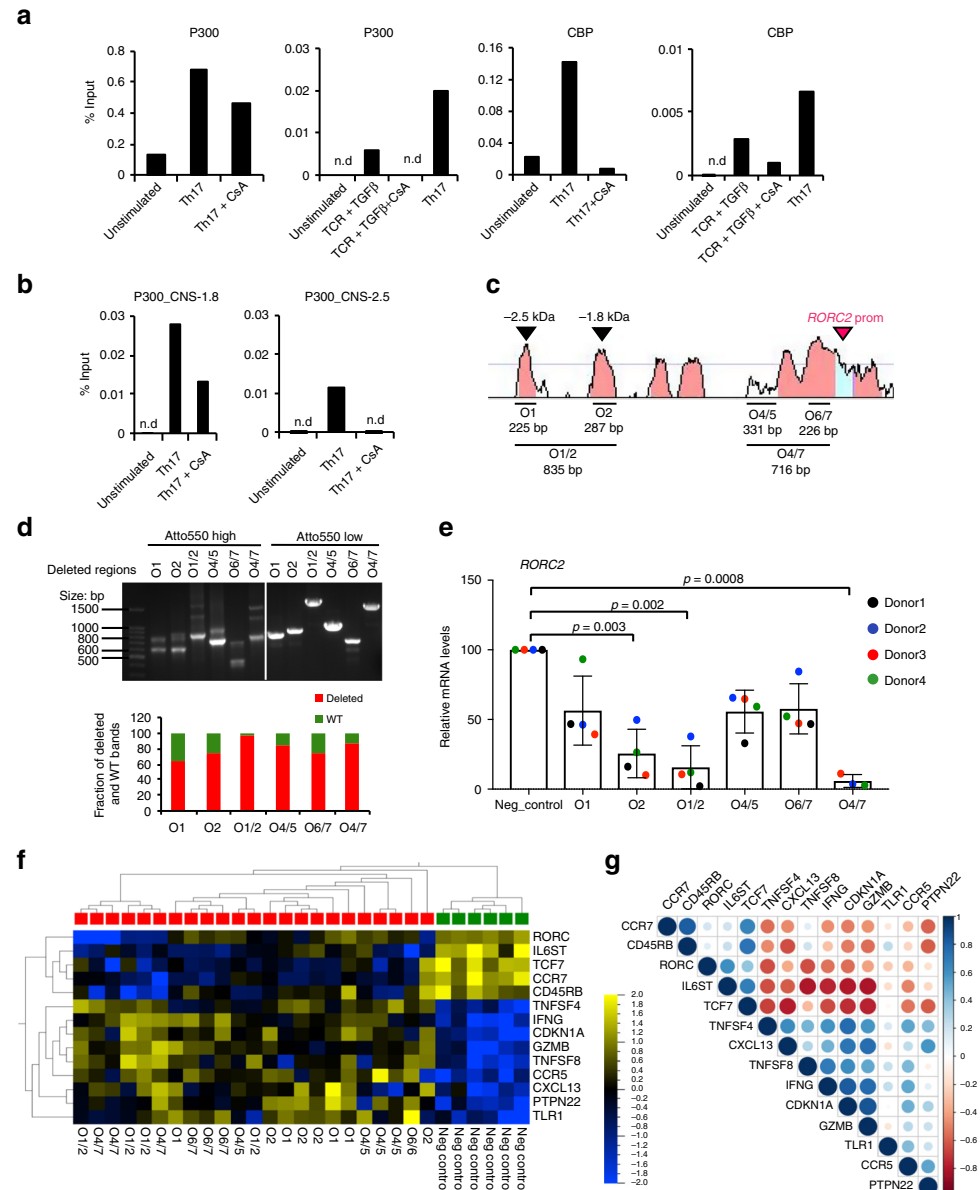

**Fig. 7** Functional characterization of RORC2 regulatory regions. **a**, **b** p300 and CBP binding to the *RORC* locus is cyclosporine-sensitive. **a** p300 and CBP ChIPs were performed in CD3[+] T cells from peripheral blood stimulated for 20 h in the shown conditions followed by RT-qPCR of the *RORC2* promoter region. Background from an irrelevant IgG was subtracted. Shown are two representative independent ChIP experiments. Additional independent experiments are shown in Supplementary Fig. 3j, k. **b** p300 ChIP was performed as in A, followed by RT-qPCR of *RORC2* enhancer regions. **c**–**g** CRISPR-Cas9-mediated deletion of regulatory regions in the *RORC* locus affects *RORC2* expression in primary CD4[+] T cells. **c** Human/murine alignment of the *RORC2* promoter and upstream conserved region (CNS-2.5 kb and −1.8 kb). The deleted regions (horizontal bars) and their sizes are represented. **d** Top panel: PCR amplification of genomic DNA from sorted samples (donor 3) transfected with a ribocomplex of Cas9 protein, crRNA and Atto550-labeled tracrRNA. The left side of the gel shows samples enriched for transfected cells (Atto550 High), the right side samples from Atto550 low cells, carrying wild-type DNA. Asterisks mark fragments carrying the expected deletions. Bottom panel: the proportion of WT and deleted DNA fragments in the Atto550 positive cells was quantified with ImageJ program. **e** Expression of *RORC2* was measured by nCounter technology in CD4[+] T cells transfected with a negative control crRNA, or cells carrying a genomic deletion as shown in **c**, **d**. Horizontal bars indicate two-tailed Student *t*-test on samples from 4 different donors. **f** ANOVA analysis of gene expression Nanostring data in control and CRISPR/Cas9 deleted CD4[+] T cell populations from 4 donors (*p* = 0.002, q = 0.060). Donors 3 and 4 had two independent control transfections each. Green squares indicate control samples, red square samples carrying *RORC2* deletions. **g** Pearson correlation plot of the top differentially expressed variable from the ANOVA analysis in **d**. Source data are provided as a Source Data file

CNS-1.8, supporting a role for NFAT in the acetylation of the enhancer region as well (Fig. 7b).

Taken together, these results indicate that the recruitment of NFAT proteins to the *RORC* locus plays a role in establishing a chromatin conformation permissive for *RORC2* transcription.

**Deletion of CNS-2.5 and CNS-1.8 reduces *RORC2* transcription.** The analysis of histone modifications and of transcriptional regulators binding suggest that the genomic regions corresponding to CNS-2.5 and CNS-1.8 may have enhancer function on *RORC2* expression. These data, together with data from the literature (Supplementary Fig. 4 and legend), indicate that these regions are

important to nucleate the assembly of clusters of transcription factors that integrate stimuli from the TCR and from cytokines to regulate *RORC*2 expression. To explore the functional significance of these regions in *RORC*2 transcription, we used CRISPR/Cas9 technology to engineer genomic deletions of *RORC*2 regulatory regions in human CD4$^+$ T lymphocytes. We deleted CNS-2.5 and CNS-1.8 individually (O1 and O2), and both in combination (O1/2), and also tested the functional importance of the region containing the O4 and O5 NFAT binding sites (Fig. 7c and Supplementary Fig. 5a, b). As a positive control we introduced a deletion of the proximal *RORC*2 promoter (O6/7), and a larger one of the extended promoter region (O4/7, Fig. 7c and Supplementary Fig. 5a). Targeting efficacy of guide RNAs was tested in Jurkat cells (Supplementary Fig. 6a).

Transfection of crRNA/Cas9 riboprotein complexes in human primary CD4$^+$ T cells achieved deletion of the targeted regions in 4 different donors, as confirmed by PCR amplification of the genomic target region (Fig. 7d and Supplementary Fig. 6b–d). We used nCounter technology (Nanostring) to quantify the effects of the genomic deletions on the expression of *RORC*2 and of a panel of immune-related genes. Analysis of *RORC*2 expression showed significantly reduced *RORC*2 transcripts in cells carrying deletions of the *RORC* regulatory regions, both at the promoter and at the enhancer, compared to cells transfected with the negative control ribocomplex (Fig. 7e). Contemporary deletion of CNS-2.5 and CNS-1.8 resulted in increased *RORC*2 inhibition, arguing for an additive function of these regions. Deletion of the region containing NFAT sites O4/5 also resulted in decreased levels of *RORC*2 transcripts, suggesting that this poorly conserved region may also play a role in human *RORC*2 regulation (Fig. 7c, e).

Multigroup comparison of gene expression in the control and samples carrying genomic deletions showed that a number of other immune genes were affected by the deletions and reduced *RORC*2 expression (Fig. 7f, Supplementary Fig. 7). In particular, we noticed an elevated IFNγ expression in CRISPR/Cas9-deleted samples, possibly due to stimulation by IL-23 present in the Th17-inducing cytokine cocktail, and activation of STAT4[34]. Also increased in samples with deletions at the *RORC* locus were transcripts from a number of IFNγ target genes, or genes associated with IFNγ-producing cells (*PTPN22*, *TNFSF8*/CD153, *CDKN1A*/p21Cip, *CCR5*, *GZMB*)[35–37], suggesting a skewing towards a Th1-type response in cells stimulated in Th17-inducing conditions when RORγt expression is inhibited. *PTPN22* was also described as a direct target of RORγt[38]. These findings support an enhancer role for CNS-2.5 and CNS-1.8.

**NF-kB cooperates with NFAT to activate *RORC*2 transcription**. NFAT proteins share the same core DNA binding site with transcription factors of the NF-kB family[30], and the NF-kB subunits c-Rel and p65 were shown to bind to the murine *RORC* and *RORC*2 promoters in CD4$^+$ T cells cultured in Th17 polarizing conditions[22].

We therefore asked whether NF-kB proteins could bind to the same genomic sites to which NFAT bound in vitro. Probing the membrane from our DNA affinity capture assay with anti-p65 or anti-p50 antibodies showed that both proteins were capable of binding to three of the NFAT-binding oligos in the *RORC*2 promoter (sites 4, 6 and 7, Fig. 8a). No binding was detected at the NFAT sites between the two promoters (oligos 1, 2 and 3, Fig. 8a).

ChIP experiments in CD4$^+$ cord blood T cells with an antibody against the p65 NF-kB subunit confirmed binding of NF-kB to the *RORC*2 promoter, as well as to upstream enhancer (CNS-1.8 Fig. 8b). Notably, p65 binding was detectable in cells stimulated in Th17 conditions, but not in naive unstimulated

cells, nor in TCR-stimulated cells in the presence of TGFβ alone. These data indicate that different transcriptional complexes assemble at the *RORC*2 promoter in different stimulation conditions, and that Th17-inducing cytokines are important for NF-kB binding to the *RORC*2 locus and full activation of human *RORC*2 expression.

To test the functional effect of NF-kB transcription factors on the *RORC*2 promoter, we used reporter gene assays in HEK293T cells transfected with the *RORC*2 promoter reporter construct. Overexpression of both p65 and p50 (NFkB, Fig. 8c) significantly increased transcriptional activity relative to the activity of the *RORC*2 promoter alone (first bar in the plots of Fig. 8c). While co-transfection of NFATc1A or NFATc1C did not further increase transcriptional activity compared to what observed with NF-kB alone, co-expression of NFATc2 with NF-kB had an additive effect on promoter activity, indicating that these transcription factors cooperate in inducing *RORC*2 expression (Fig. 8c). As in the case of NFAT, NF-kB overexpression did not increase luciferase activity promoted by the construct containing CNS-2.5 kb and −1.8 kb, when compared to the construct containing only the *RORC*2 promoter (Supplementary Fig. 3h), indicating that binding of NF-kB in this region is not sufficient to further increase *RORC*2 transcription, at least in a non-chromatin context.

**NFAT binds to the *RORC* locus in thymocytes**. Our epigenetic analysis of the *RORC* locus in thymocytes and in differentiating Th cells revealed an overlap in the regions that undergo histone modifications upon *RORC*2 expression. We therefore investigated whether NFAT family transcription factors could be implicated in *RORC* locus regulation in the thymus as well. NFATc3 is the main isoform expressed in DP thymocytes[39], and ectopic NFATc3 expression in HEK293T cells could induce a modest but statistically significant increase in *RORC*2 promoter transcriptional activity (Fig. 8d). We therefore asked whether NFATc3 factors bound to the *RORC* locus in vivo. ChIP assays demonstrated that in the human thymus NFATc3 is bound to the same *RORC*2 promoter and enhancer regions that were bound by NFATc2 and NFATc1 in activated CD4$^+$ T cells in the periphery (Fig. 8e). However, we observed similar levels of NFATc3 binding in DP and in SPCD4$^+$ thymocytes, despite the marked differences in *RORC*2 transcription in the two populations (Figs. 1b and 8e). NFATc1 was also shown to play a role in the early developmental stages of thymocytes[40]. We found that also NFATc1 bound to the *RORC* locus in both DP and SPCD4$^+$ thymocytes (Fig. 8e). Taken together with the findings that *RORC* locus acetylation persists in SPCD4$^+$ cells despite downregulation of *RORC*2 expression, these data are compatible with an indirect regulatory role for NFAT proteins in thymocytes in the maintenance of a permissive chromatin environment for *RORC*2 expression.

**Discussion**

In this work we have analyzed the early events that guide the expression of RORγt in human CD4$^+$ T cells. Our data show that the calcineurin/NFAT pathway, and in particular NFATc2, plays a direct role in controlling RORγt transcription, by binding to the *RORC* locus and promoting a permissive chromatin conformation at *RORC*2 regulatory regions.

Murine models have shown that NFAT is essential in directing the development of CD4$^+$ T cell subsets. The main NFAT factors expressed in peripheral T lymphocytes are NFATc1 and NFATc2. While NFATc1 has also a role in thymocyte development[40], NFATc2 plays a role mainly in the periphery, since *NFATc2*$^{-/-}$ mice show normal thymus development but alterations in Th2 differentiation[41]. Simultaneous deletion of NFATc1 and NFATc2 in

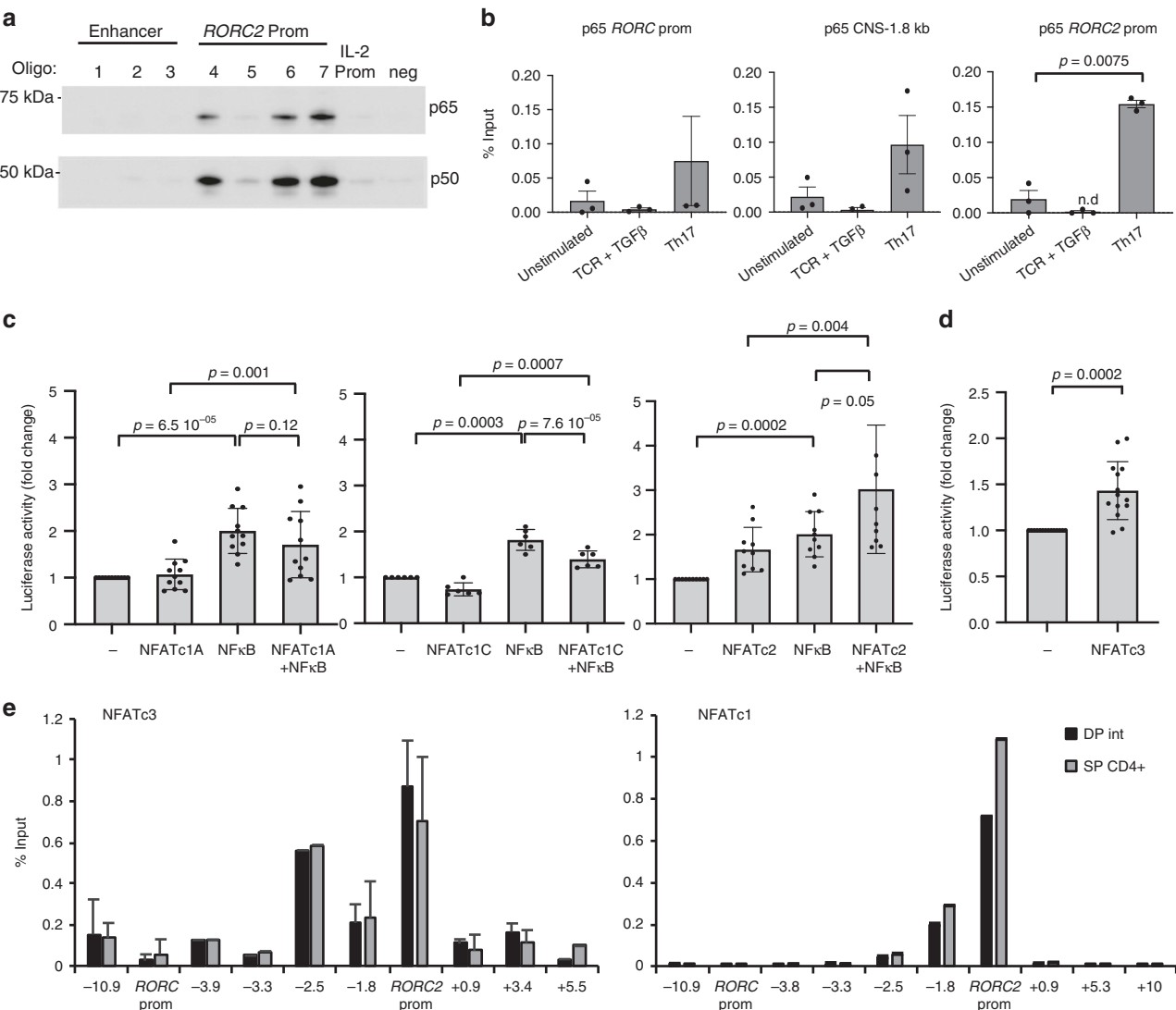

**Fig. 8** NF-kB cooperates with NFAT in activating *RORC2* transcription. **a** NF-kB proteins bind to the *RORC2* promoter in vitro. DNA affinity capture assay was performed in Jurkat E6.1 cells with biotinylated oligos encompassing the NFAT binding sites (see Fig. 5). Western blot of the DNA-bound proteins was probed with anti-p65 and anti-p50 antibodies. *IL-2 prom*: NFAT binding site from the *IL-2* promoter, *neg*: sequence in the *RORC* locus with no NFAT binding site. **b** NF-kB binds to the human *RORC* locus in stimulated CD4$^+$ T cells in vivo. CD4$^+$ T cells from cord blood were stimulated for 20 h in the indicated conditions. ChIP was performed with an anti-p65 antibody, followed by RT-qPCR of the *RORC* promoters and CNS-1.8 kb enhancer region. Data are shown as percent of input, after subtraction of the irrelevant IgG background and represent average and SD of 3 experiments. n.d.: not detected. **c** NF-kB cooperates with NFAT to activate transcription from the human *RORC2* promoter. HEK293T cells were transfected with expression vectors for p65 and p50 (NF-kB) or the indicated NFAT transcription factors, together with a *RORC2* promoter construct. Normalized transcriptional activity is represented as fold changes over the activity of the *RORC2* promoter alone (first bars). Data shown represent average and SD of 11, 6 and 10 independent experiments (for left, middle and right graphs respectively). **d** NFATc3 activates transcription from the *RORC2* promoter. HEK293T cells were transfected with the *RORC2* promoter in the presence or absence of NFATc3. Luciferase activity was measured after 20 h and represented as fold change relative to activity of the *RORC2* promoter alone (first bar). Plotted are average and SD from 14 transfections. **e** NFATc3 binds to the *RORC* locus. Extracts from sorted DPint and SP CD4$^+$ thymocytes were subjected to chromatin immunoprecipitation with anti-NFATc3 antibodies (left panel, average and SD of 2–4 replicates for the different regions) and anti-NFATc1 antibodies (right panel, one representative experiment). NFAT enrichment is represented as percentage of input after background subtraction. Shown are the p-values from two-tailed *t*-tests performed between the indicated conditions. Source data are provided as a Source Data file

CD4$^+$ T cells leads to a broad suppression of cytokine secretion, demonstrating that these factors are essential for Th effector differentiation and function[42]. These studies demonstrated that the deficiency of individual NFAT proteins results in distinct phenotypes, implying that different NFAT factors can display unique transcriptional activities, and that the balance between NFAT proteins may be crucial to sustain specific CD4$^+$ T cell functions. However, dissecting the mechanism of action of single NFAT factors has proven challenging, due to partially overlapping activities of the different NFAT proteins, and their complex interplay. This includes feedforward loops, such as NFATc1 induction by NFATc2[43], and compensatory mechanisms in knock-out models, which have complicated the study of NFAT role in murine Th17 function. NFATc1-deficiency in CD4$^+$ T cells causes reduced levels of Th17 cytokines secretion, and attenuated experimental autoimmune encephalitis (EAE)[44,45]. Furthermore, the combined deficiency of NFATc1 and NFATc2 completely protects mice from EAE, suggesting that both factors are involved in Th17 differentiation or function, and in the production of IL-17[44,46,47]. NFATc1 deficiency alone resulted in a modest reduction of *RORC*

expression, but unexpectedly NFATc2$^{-/-}$ CD4$^+$ T cells in Th17-inducing conditions showed increased expression of *RORC*. These results may be explained by the compensatory action of increased nuclear NFATc1 activity observed in *NFATc2*$^{-/-}$ animals[44], and are consistent with the ability of NFATc1 to bind to the NFAT sites at the *RORC* locus, in vitro or under specific conditions in vivo. In contrast with the NFATc2 ko findings, a hyperactive form of NFATc2 induced early upregulation of *RORC* transcription[47], and consistent with this study and with the findings of Kim et al.[31], we found that NFATc2, but not NFATc1, bound to the human *RORC* locus in early differentiating Th17 cells. NFATc2, but not NFATc1, synergized with NF-kB factors to activate the *RORC*2 promoter, supporting a more important role for this NFAT family member in the early induction of *RORC*2 transcription in human Th17 cells. The preference for NFATc2 binding in Th17 cells could be explained by the different kinetics of NFATc1 and NFATc2 in these conditions, since NFATc2 showed strong nuclear accumulation in stimulated cord-blood CD4$^+$ T cells early after activation. Another possibility, is that preferential binding of NFATc2 to the *RORC*2 locus in differentiating Th17 cells is stabilized by the interaction with other transcription factors induced in Th17 conditions, consistent with our finding that NF-kB p65 binds to the *RORC*2 promoter only in Th17 differentiating conditions. A recent mass-spectrometric analysis of NFATc1- and NFATc2-binding proteins in Jurkat T cells has demonstrated that these factors can establish non-overlapping interactions with other proteins, including preferential interactions of NFATc2 with some transcriptional regulators, such as c-Jun, c-Fos, Runx1, EGR1 and the MafG and K transcription factors[48]. These findings support the notion that NFAT factors act through the nucleation of multi-protein complexes.

Our luciferase reporter assays in HEK293T cells demonstrate that NFAT factors can cooperate with NF-kB to activate the *RORC*2 promoter, although the increase in transcriptional activity in the presence of NFAT proteins was moderate. This could be explained by the lack, in our experimental conditions, of additional factors necessary for full transcriptional activation of *RORC*2. It is also possible that an important function of NFAT binding to the *RORC*2 locus is to promote the establishment of a permissive chromatin conformation. This activity cannot be appreciated in the non-chromatin environment of the reporter assay, but is supported by the fact that CsA inhibits H4Ac and H3K27Ac, and binding of histone acetylases at the *RORC* locus.

Our analysis of histone modifications in developing thymocytes and differentiating Th17 cells identified in the genomic sequence between *RORC* and *RORC*2 transcription start sites several conserved regions with possible regulatory function, which may be important to cluster transcription factors that integrate stimuli from the TCR and from cytokines to regulate *RORC*2 expression (Supplementary Fig. 4). NFAT presence at these regions could be essential to promote the binding of such transcriptional complexes, by translating TCR signaling in permissive chromatin remodeling of selected regulatory sequences. CRISPR/Cas9-mediated deletion of these sequences confirmed their role in *RORC*2 expression, and resulted in increased expression of genes associated with a Th1 response, similarly to what observed in murine Th17 cells treated with RORgt antagonists[38].

The *RORC*2 promoter (Supplementary Fig. 4a) presents multiple NFAT-consensus binding sites that showed strong binding of NFAT proteins in vitro and in vivo. Sequences that can bind both NFAT and NF-kB were described in several cytokine promoters[29]. Consistently, we found that NF-kB proteins also bind to three NFAT-binding sequences in the *RORC*2 promoter, and that NF-kB and NFAT cooperate in promoting *RORC*2 transcription. These data are consistent with the role of c-Rel and p65 in

activating the murine Rorγt promoter reported by Ruan et al.[22]. One of the two Rel sites described by these authors is strictly conserved between mouse and human, and partially overlaps with the NFAT site 7 of our study. NFAT site 7 is also adjacent to a Runx3 site that was described to promote Rorγt transcription in NK cells and ILC3[49].

NFAT sites 6 and 7 are in the proximity of two predicted Hif-1α binding sites. Hif-1α is upregulated in a STAT3-dependent manner and binds to this region in murine Th17 cells[20]. This sequence may thus constitute one of the molecular nodes that integrate environmental clues with cytokine and TCR signaling.

The upstream conserved region (CNS-2.5/-1.8 kb, Supplementary Fig. 4b) bears markers of active enhancers in cell populations that express *RORC*2. The *RORC*2 enhancer region is characterized by weaker binding of NFAT to sites 1 and 2 in vitro, which may explain the lack of enhancing activity of this fragment in our reporter assay, since binding of additional factors at these sites may be necessary to stabilize the interaction of NFAT proteins in vivo.

Data from the literature indicate that two clusters of transcription factors can bind to this region in Th17 cells. Runx1 and E-proteins have been described to bind in the proximity of NFAT site 1[24,50,51]. The second cluster is found at CNS-1.8, which encompasses NFAT site 2. This sequence was shown to bind c-Maf and Sox5t in murine CD4$^+$ T cells[21]. An interaction between c-Maf and NFATc2 was demonstrated to maximally activate the *IL-4* promoter[52]. Given the close proximity of NFAT site 2 with a c-Maf site, and the ability of c-Maf to activate or repress transcription in Th17 cells[53], it will be interesting to explore how NFATc2 and c-Maf functionally interact to regulate *RORC*2 transcription.

Our epigenetic analysis also highlighted the presence of enhancer regulatory features in the first *RORC*2 intron (CNS+5.5), which does not bind NFAT in our study. Binding of BATF/IRF4 complexes, STAT3 and p300 was reported in this region in murine CD4$^+$ T cells. Ciofani et al. proposed that BATF4/IRF4 complexes act as pioneer factors to promote chromatin accessibility for the binding of additional cytokine-induced regulators[53]. The BATF/IRF4 sites in the *RORC*2 intron may therefore represent another NFAT-independent integration node for TCR/cytokine signaling for the regulation of *RORC* expression. Since *Batf*$^{-/-}$ mice showed normal early *RORC* induction, but could not maintain *RORC* levels upon longer stimulation, this region may have a role in stabilizing *RORC* transcription at later time points after stimulation[54]. In our study, this *RORC*2 intron showed marked H4Ac and H3K27Ac enrichment in DP thymocytes. However, at early time points during in vitro Th17 differentiation, increased acetylation was limited in this region, consistent with a lack of early involvement of this intron in *RORC*2 induction.

Whether the factors that play a role in *RORC*/*RORC*2 expression in Th17 cells also are important for thymic *RORC*/*RORC*2 induction remains to be established. However, the histone modifications in the sequences upstream the *RORC*2 TSS showed remarkable similarities in *RORC*2-expressing thymocytes and differentiating Th17 cells, suggesting conserved regulatory functions in the thymus and in the periphery. The integrity of the calcineurin/NFAT pathway is essential for normal thymic development[55]. NFATc3 is the major NFAT protein expressed in thymocytes[39], and NFATc3 deficient thymocytes present developmental defects both in the late DN stage, and at the DP stage[56]. The binding of NFATc3 to the *RORC*2 locus that we detected in thymocytes indicates that one of NFATc3 functions in these cells may be to regulate *RORC*2 transcription. Binding of NFAT at the *RORC*2 promoter and enhancer could still be detected in SP CD4$^+$ thymocytes, which do not express *RORC*2, demonstrating that NFAT presence is not sufficient for *RORC*2 expression,

although it may be important to conserve low levels of H4Ac and a permissive conformation of the *RORC* locus for rapid activation in the periphery.

Epigenetic analyses of Th cell differentiation are consistent with the notion that commitment to a stable Th state is supported by chromatin remodeling at genes important for the effector phenotype, and that the accessible conformation can persist in the absence of the initial inducing signal and of active transcription[57]. The coexistence of "bivalent" (permissive H3K4me3 and repressive H3K27me3) modifications at the *Tbx21* and *Gata3* loci in naive CD4[+] T cells may reflect their previous activation in the thymus[8]. We found that the *RORC* locus also acquires a "primed" configuration during thymic development, with diffused histone acetylation, reduced H3K27me3 and widespread enrichment of H3K4me1. Memory of *RORC*2 expression persists in mature naive CD4[+] T lymphocytes, which show a partially permissive conformation at the *RORC*2 locus, with reestablished H3K27me3 together with H4Ac. Low levels of NFATc2 binding in naive peripheral CD4[+] T cells may contribute to maintaining the acetylated status of the locus, in the absence of *RORC*2 expression.

In modeling the epigenetic differentiation of Th2 cells, Agarwal and Rao proposed a two-step process for the activation of the IL-4 enhancer, where subset-specific transcription factors establish an open structure at the *IL4* locus, making it accessible to non-subset specific factors that are induced by antigen-stimulation[58]. In the case of the *RORC* locus, we propose that TCR-mediated activation of non-specific factors, such as NFAT, is the priming event resulting in an accessible chromatin conformation of this locus and low-level transcription. The final outcome for Th differentiation depends on the stabilization (or repression) of the activating modifications through the binding to the open chromatin of additional factors induced by polarizing cytokines, resulting in tissue-specific transcription.

## Methods

**Primary cells and cell lines**. Human thymi were removed during cardiac surgery at the Pediatric Cardiac Surgery Department of Hôpital Necker, Paris, France. Both parents provided a declaration of non-opposition to the secondary utilization, and the study was approved by the local ethics committee (Comité du Recherche Clinique, CoRC, Institut Pasteur, Paris, no. 2011-43). Thymi were dissected in small fragments, passed through a sieve and the cellular suspension was separated on Ficoll-Hypaque (GE Healthcare) density gradient centrifugation. Thymocytes subpopulations were isolated by fluorescence-activated cell sorting on a FACS Aria II (BD Biosciences). DAPI and a "dump channel" with antibodies against CD56, CD19, CD14, CD11c, TCRγ/δ, and glycophorin were used to eliminate dead cells, NK cells, B cells, monocytes, dendritic cells, γ/δT cells and erythrocytes. The remaining population was sorted into Double Negative (DN, CD4[−]CD8[−]), CD8 Single Positive (SP CD8, CD4[−]CD8[+]), and Double Positive cells (DP, CD4[+]CD8[+]) cells. CD4[+]CD8[−] cells were separated into Immature Single Positive (ISP, CD4[+]CD8[−]CD3[−]) and CD4 Single Positive (SP CD4, CD4[+]CD8[−]CD3[++]). The DP population was separated into an immature population with low CD3 levels (DP CD3low) and a more mature population with intermediate levels of CD3 expression (DP CD3int) (Supplementary Fig. 1a for gating strategy).

Cord Blood was obtained from the AP-HP Cord Blood Bank (Hôpital Saint Louis, Paris, France), as units discarded following screening for allogeneic transplantation. Written informed consent was obtained from the donors. The research protocol was authorized by the CoRC, Institut Pasteur, Paris (no. 2012-38) and the regional (CPP 2013/01NICB, Comité de Protéction des Personnes) ethical committees. Lymphocytes were isolated by Ficoll-Hypaque density gradient centrifugation, and CD4[+] or CD8[+] population by either fluorescence-activated or magnetic cell sorting (Miltenyi Biotech), following the manufacturer's protocol for positive selection.

CD3[+] and CD4[+] T cells from adult peripheral blood were isolated from buffy-coats (Hôpital Saint Louis, Paris, France) by Ficoll-Hypaque density gradient centrifugation, followed by positive selection on magnetic columns using the CD3 or CD4 microbeads according to the manufacturer's protocol (Miltenyi Biotech).

Jurkat E6.1 cells (ATCC, TIB-152) were cultured in RPMI supplemented with 10% FCS, HEK293T cells (ATCC CRL-3216) were cultured in DMEM plus 10% FCS.

**T cells stimulation**. Purified naive CD4[+] T cells were resuspended in RPMI medium supplemented with 10% FCS and cultured in the following conditions:

**TCR**: stimulation with Dynabeads Human T-activator CD3/CD28 (4 beads per 10[6] cells; Thermo Fisher scientific); **TCR + TGFβ**: TCR stimulation in the presence of TGFβ (10 ng/ml; Miltenyi); **Th17** polarizing condition: TCR stimulation in the presence of TGFβ (10 ng/ml), IL-1β (10 ng/ml; Miltenyi), IL-23 (10 ng/ml; Miltenyi) and IL-21 (25 ng/ml; Miltenyi), plus anti-IFNγ and anti-IL-4 neutralizing antibodies (1 μg/ml; BD Biosciences); **Th1** polarizing condition: TCR stimulation in the presence of IL-12 (2.5 ng/ml; Roche) and anti-IL-4 antibodies. The calcineurin inhibitor Cyclosporine A (CsA; 1 μg/ml; Sigma-Aldrich) was added to the cultures where indicated.

**Gene expression analysis**. Total RNA was purified on RNeasy columns (Qiagen) and reverse transcribed using the High Capacity cDNA Reverse Transcription kit (Applied Biosystems). Real-Time quantitative PCR was performed on a Step One Plus PCR System (Applied Biosystems), with SYBR green FastStart Master Mix (Roche) and the data were normalized to 18S content. The primers used are detailed in Supplementary Table 2.

**Imaging Flow Cytometry**. CD4[+] T cells isolated from cord blood were stimulated as indicated, and fixed in 4% paraformaldehyde at RT for 10 min. Cells were washed once in FACS buffer (PBS, 2% FCS) and resuspended in permeabilization buffer (10% Triton-X100 in FACS buffer) with mouse or rabbit normal IgG for 10 min, RT, followed by incubation at RT for 30 min with anti-NFAT or irrelevant control antibodies (see Supplementary Table 7). Samples were washed once with FACS buffer and incubated with anti-mouse IgG antibodies and DAPI for 30 min at RT. Data acquisition was performed on an ImageStreamX MKII Imaging cytometer (Amnis, Luminex corporation) and data processed using the IDEAS software (Amnis). Images acquired include a brightfield image (Channels 1 and 9; 430–480 nm), Cyan3 (Channel 2; 560–595 nm), DAPI (Channel 7; 430–505 nm) and AF647 (Channel 11; 660–740 nm). Cyan3 was excited by a 56 nm laser with a 150 mW output, DAPI was excited by a 405 nm laser with a 1 mW output, and AF647 was excited by a 642 nm laser with a 150 mW output. For each sample, images were collected for 10,000 events. A matrix of spectral compensation values was built using values from unstained cells, and samples labeled with a single fluorochrome. Following compensation for spectral overlap, nuclear translocation was computed with IDEAS® software version 6.2, using the Similarity feature, which is derived from the logarithmic transformation of the Pearson's correlation coefficient calculated between pairs of the intensity values obtained from the NFAT image and the nuclear reference (DAPI) image[59,60]. Single cells that were in focus and were positive for DAPI were selected and all measurements of NFATc1 and NFATc2 were made from this Single cells/DAPI positive population (gating strategy in Supplementary Fig. 2d, and antibodies list in Supplementary Table 8).

**Chromatin Immunoprecipitation assay**. After stimulation, cells were fixed in 1% formaldehyde for 10 min at 37 °C. The reaction was stopped by adding glycine to a final concentration of 0.125 M. Cells were lysed on ice in SDS lysis buffer (1% SDS, 10 mMEDTA, and 50 mM Tris, pH8.0, AEBSF (Roche) and Leupeptin (Sigma-Aldrich)). We used 1 million cells for histone modification, 8 million cells for NFAT/NFκB and 25 million cells for the p300/CBP ChIPs. Cells were sonicated using a Bioruptor-Pico (Diagenode) for 15 cycles. The sonicated lysate was diluted 10-fold in ChIP dilution buffer (0.01% SDS, 1.1% Triton X-100, 2 mM EDTA, 20 mM Tris–HCl, 500 mM NaCl). In total 10% of the diluted lysate was kept as "input" sample for normalization. Samples were incubated overnight at 4 °C with the antibody, and the antibody/protein complex was collected with Dynabeads protein A or protein G (Thermo Fisher scientific). Antibodies used and concentrations are detailed in Supplementary Table 5. The Ab/protein complex was washed with Low-salt Buffer (0.1% SDS, 1% Triton X-100, 2 mM EDTA, 20 mM Tris–HCl, pH 8.0, 150 mM NaCl), High-salt Buffer (0.1% SDS, 1% Triton X-100, 2 mM EDTA, 20 mM Tris–HCl, 500 mM NaCl), LiCl Buffer (0.25 mM LiCl, 1% IGEPAL-CA630, 1% deoxycholic acid, 1 mM EDTA, 10 mM Tris, pH8.0), and twice with TE Buffer. DNA/protein complexes were eluted with 1% SDS and 0.1 mM NaHCO₃, and crosslinking reversed by incubation at 65 °C overnight, followed by Proteinase K digestion. DNA was purified using the QIAquick PCR purification kit (Qiagen) and analyzed by RT-qPCR. Primers are listed in Supplementary Table 1. Data were plotted after subtraction of Irrelevant IgG background, and eventual negative values were indicated as non detected.

**Affinity capture assay for DNA binding proteins**. NFAT binding sites were predicted using the PROMO 3.0 program. Biotinylated forward-strand and a reserve-strand oligonucleotides encompassing the predicted binding site plus 10 bases upstream and downstream were obtained from MWG Biotech (Supplementary Table 3). Nuclear extracts were prepared from $5 \times 10^6$ Jurkat E6.1 or primary CD4[+] T cells (extraction buffer: 20 mM Hepes/NaOH pH 7.9, 0.35 M NaCl, 0.1% Nonidet P-40, 1 mM MgCl₂, 0.5 mM EDTA, 20% glycerol and proteinase inhibitors) and incubated for 30 min on ice with 2 pmoles of biotinylated double-stranded oligonucleotides and 2 μg poly(dI-dC). Bound proteins were pulled down with 20 μl streptavidin/magnetic dynabeads (Thermo Fisher Scientific) and washed 4 times with ice-cold wash buffer (25 mM Hepes/NaOH pH 7.9, 0.15 M NaCl, 0.1% Nonidet P-40) and once with ice-cold PBS. The complexes beads/oligonucleotides/proteins were eluted in 2× sample buffer at 95 °C for 5 min.

Proteins were analyzed by SDS gel and western Blot with anti-NFATc1, anti-NFATc2, anti-p50 and anti-p65 antibodies (Supplementary Table 5). Uncropped original blots are provided in a separate Source File.

**Generation of RORC reporter gene constructs.** To measure the activity of the RORC2 promoter in vitro, an 832 bp fragment (−715 to +117 from the RORC2 transcription start site) was amplified from a RORC Bacterial Artificial chromosome (BAC CTD 2234L24; Invitrogen) and cloned in the pGL3 basic plasmid (Promega). A RORC regulatory element located between the RORC and RORC2 promoters (CNS-2.5/-1.8, that contains two CNS) was also amplified and cloned upstream of RORC2prom in the pGL3 vector. The primers used to generate these constructs are listed in Supplementary Table 4.

**Cell transfection.** Jurkat E6.1 cells were transfected using the gene Pulse Xcell electroporator (Bio-rad, 260 V, 950 µF) with 500 ng of pGL3 reporter vectors and 5 ng of pRL-TK vector (Renilla luciferase, to normalize for transfection efficiency). HEK293T cells were transfected using Lipofectamine 2000 (Life Technologies). Cells were transfected in a 48 well plate with 100 ng of pGL3 reporter vectors, 5 ng of pRL-TK vector (Renilla luciferase, to normalize for transfection efficiency), and 180 ng of the following expression vectors: pEF6-NFATc2 expression vector (a kind gift from O. Kaminuma), pEGZ-NFATc1A and pEGZ-NFAT1C (a kind gift from F. Berberich-Siebelt), pCMV4-p50 (purchased from Addgene) and pCDNA-p65 (a kind gift from G. Natoli). Luciferase activity was measured 24 h after transfection using the Dual-Luciferase Reporter Assay System kit (Promega).

**CRISPR/Cas9 editing of Jurkat cells.** Cas9 was expressed from plasmid pCas9-GFP (Addgene plasmid #44719) and guide RNAs (gRNAs) were expressed from plasmid MLM3636 (Addgene plasmid # 43860). The choice of gRNA was based on high target specificity and low number of off-target sites, as determined with the Benchling and Cas OFFinder softwares (see Supplementary Table 6). Two gRNA-related oligonucleotides (upstream and downstream) containing the 20 bp gRNA sequence and the linker sequences (5′-ACACCG-gRNA-G-3′ for the upstream oligonucleotide and 5′-AAAAC-gRNA′-CG-3′ for the downstream oligonucleotide) were used for cloning into the BsmBI cloning site of MLM3636[61]. Briefly, annealing of gRNA-related upstream and downstream oligonucleotides (100 µM stock) was performed by mixing 10 µl of each oligonucleotide (Final oligo concentration: 10 µM) with 80 µl of buffer containing 10 mM Tris pH7,5, 1 mM EDTA and 50 mM NaCl and incubating at 95 °C for 15 min and then at room temperature for at least 1 h. MLM3636 plasmid was digested by BsmBI enzyme (NEB). Linearized MLM3636 plasmid was run on a 1% gel, extracted and purified (Qiagen, #28704). Ligation was performed by incubating 100 ng of linearized MLM3636 plasmid with 0.05 µM annealed gRNA-related oligonucleotides, 2 µl 10× Ligase buffer (NEB) and 1 µl T4 Ligase (NEB) in a 20 µl volume reaction for 1 h at room temperature. DH5α bacteria (Invitrogen) were transformed with the ligation product and amplified. Purified plasmids were Sanger sequenced using LKO.1 5′ primer (5′-GACTATCATATGCTTACCGT-3′) to verify correct gRNA cloning. 5 million Jurkat cells were nucleofected using the Cell Line Nucleofector® Kit V from Lonza (program X-001, Amaxa Nucleofector Technology) and 4 µg of upstream gRNA, 4 µg downstream gRNA and 4 µg of the pCas9-GFP plasmid. The following day, single cells expressing GFP were sorted using a BD FACSAria™ cell sorter. Plates containing nucleofected cells were then kept in culture four to six weeks, and genomic DNA from growing cell clones were screened by PCR for deletions (primers listed in Supplementary Table 7).

**CRISPR/Cas9 editing of primary CD4+ T cells.** crRNAs for the sequences validated in Jurkat cells were obtained from Integrated DNA technologies (IDT, Skokie, IL, USA). TracrRNA ATTO550, S.p Cas9 Nuclease V3, non-targeting negative control crRNA, and the Cas9 electroporation Enhancer were purchased from IDT. Guide RNA (gRNA) complexes were formed by heating 1 µl 200-µM crRNA, 1 µl 200-µM tracrRNA and 2.5 µl of Duplex buffer at 95 °C for 5 min, and incubating at room temperature for 15 min. To form RNP complexes, 4 µl of upstream gRNA, 4 µl of downstream gRNA, and 5 µl Cas9 nuclease (36 µM) were incubated at room temperature for 20 min. For RNP electroporation, 5 million CD4+ T cells freshly isolated from healthy donor were stimulated 48 h through the TCR using αCD3/αCD28 beads (4 beads per $10^6$ cells; Thermo Fisher scientific). Cells were resuspended in 87 µl T buffer (Invitrogen) with 13 µl RNP complex and 20 µl 10.8 µM Cas9 electroporation enhancer. Electroporation was performed at 1600 V 3 pulses of 10 ms with the Neon Electroporation system (Invitrogen). In total 24 h post electroporation, the top 15–20% ATTO550 high and the ATTO550 low cells were sorted, and plated in culture medium supplemented with Th17 polarizing cocktail and cultured for 4 days before analysis. RNA and genomic DNA were isolated using the AllPrep DNA/RNA Micro Kit (Qiagen RNeasy Micro Kit, Valencia, CA). Genomic DNA from the sorted cells was analyzed using the primers listed in Supplementary Table 7 and the Expand long template PCR system (Roche). RNA concentration was estimated using Qubit RNA HS Assay Kit (Life Technologies, USA) according to the manufacturer's instructions and gene expression was analyzed using the nanoString Human Immunology V2 codeset, which includes 594 immune-related gene probes (nanoString Technologies), 8

negative control probes and 6 positive control probes designed against six in vitro transcribed RNA targets at a range of concentrations.

**Gene expression analysis with nCounter technology.** In total 50 ng of total RNA from each sample were analyzed according to manufacturer's instructions. Briefly, hybridization reactions were performed at 65 °C for 22 h with 5 µl RNA hybridization buffer, Reporter probes and Capture probes. Cartridges were read on a nCounter Digital Analyzer at the highest resolution (555 fields-of-view (FOV) collected per flow cell) to yield a Reporter Code Count (RCC) data set. nSolver analysis software (version 3.0, NanoString) was used for quality control and data normalization. The following housekeeping genes were used to normalize the time course experiment samples: ABCF1, EEF1G, POLR2A, RPL19 and TBP. For the CRISPR/Cas9 experiment, ABCF1, EEF1G, POLR2A, RPL19, TBP, PPIA, OAZ1 and GUSB were used. Were removed from the analysis: probes with low counts in 100% of the samples (counts below the mean of the negative control probes +2 SD in all samples), probes mapping to multiple genes or probes aligning to polymorphic regions with greater than two SNPs[62] and probes used for housekeeping gene normalization, leaving a total of 356 genes. Data analysis was performed using the Qlucore software (version 3.5, Qlucore, Lund, Sweden) was used for ANOVA statistical analysis of log2 transformed data. For Heatmap and box-plot representations, data were mean-centered, and scaled to unit variance. Rstudio (Version 1.2.1335) and the package Corrplot were used for correlation analyses. Dot-plot/column graphs were compiled with GraphPad Prism v.7.0.

**Reporting summary.** Further information on research design is available in the Nature Research Reporting Summary linked to this article.

## Data availability
Original Image Flow Cytometry files are available from the authors upon request. The source data underlying Figs. 1, 2, 3, 5, 6, 7, 8 and Supplementary Figs. 1a–d, 3, 6 and 7 are provided as a separate Source Data file.

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

## Acknowledgements

Cell separation was conducted in the Center for Translational Research/Cytometry Biomarkers Unit of Technology and Service (CRT/CB UTechS), Institut Pasteur. Estelle Mottez (CRT/CB UTechS) assisted with the drafting of clinical protocols. We thank O. Kaminuma, F. Berberich-Siebelt and G. Natoli for generously providing expression constructs (see Methods). We are grateful to Hélène Strick-Marchand (Innate Immunity Unit, Institut Pasteur) for human hepatocytes lysates. This study was supported by grants from Institut Pasteur, FOREUM Foundation for Research in Rheumatology, the Fondation Arthritis, MSD Avenir (Project iCARE-SpA), and a Bourse Passerelle from Pfizer.

## Author contributions

H.Y.C. designed and performed experiments, analyzed data, wrote the manuscript; M.R., D.L., T.S. and K.P. performed experiments; C.L. provided reagents, protocols and discussion for CRISPR/Cas9 experiments; J.L. provided key reagents; L.R. designed experiments and analyzed data, E.B. designed and performed experiments, analyzed data, wrote the manuscript.

## Competing interests

The authors declare no competing interests
