## [Peer Review File · Nature Communications]

Reviewers' comments:

Reviewer #1 (T17, TF regulation, RORgt)(Remarks to the Author):

This manuscript by Hanane et al described the function of NFAT in the regulation of expression of a functionally important transcriptional factor RORgt in Th17 cells. However, a paper by Kyun-Do Kim (Journal of Immunology, 192: 119) has described the function of Ca²⁺ channel Orai1-regulated activation of NFAT in Th17 function by regulating RORgt expression, which affects the novelty of this manuscript. In addition, this paper was not mentioned in the manuscript.

With that said, this manuscript did reveal some very useful detailed information: 1) define epigenetic modification of RORgt locus; 2) identify some potential regulatory elements; 3) human Th17 cells and thymocytes. The problem with identified multiple regulatory elements is their unknown physiological function. At least, is it possible to use CRISPR/CAS9 technology to mutate some sites to determine their function in the regulation of RORgt expression.

In summary, this manuscript provides some detailed information about NFAT function in human Th17 cells. However, due to similar conclusion by previous publication, it is likely not novel enough to be published in Nature Communications. Whereas, publication of this manuscript in a more specialized journal will benefit the related community.

Reviewer #2 (NFAT, signalling)(Remarks to the Author):

The manuscript entitled "NFAT primes the human RORC locus for RORγt expression in CD4+ T cells" by Yahia-Cherbal et al. investigates the factors involved in the transcriptional regulation of Th17 lineage. Using a number of epigenetic and transcriptomic approaches they explore the complex multifactorial interplay of transcription factors and chromatin conformational changes involved in RORγt expression and the subsequent differentiation of Th17 cells. Analysis covers transcriptional and conformational changes in thymic and peripheral cells.

The article investigates a rather challenging question with an elegant demonstration of the key role played by the NFAT family of transcription factors in both transcriptional regulation and chromatin permissive conformation. An important number of experiments show specific roles for NFATc2 in both the direct transcriptional control of RORγt and the conformational opening necessary to CD4+ T cells differentiation.

Importantly, experiments are mainly performed using primary cells from thymus and cord blood. A detailed analysis of the RORC and RORC2 promoter regions is presented and gives important clues on the specific or more general mechanisms involved in the regulation.

Moreover the results give important insights on the differential impact of TCR- dependent or Th17 specific cytokines-dependent mechanisms in the activation of RORγt expression. The manuscript is sound; results are clear and well discussed. Important specific Th17 differentiation process is analysed in the periphery. The main and almost single concern of the reviewer is the relative impact of the different members of the NFAT family in this regulation. Despite a number of arguments well discussed in the discussion of the manuscript, the relative impact of TCR-NFATc1 versus Th17-NFATc2 mediated process would need further assessment in several experiments of NFATc1 binding and transcription mediated regulation in primary cells. The question is challenging due to a number of reciprocal regulations between NFATc1 and NFATc2, which is very well discussed. However the time-related effect of both factors should be addressed regarding the literature on these processes.

Reviewer requests:

In Figure 4, an important TCR- TGF β impact is shown on NFATC1 binding to RORC2 promoter while NFATC1 is not expressed at the highest levels. Th17 cytokines show more impact on NFATC2 binding. Therefore, results shown in Figure 4F would be strengthened by the parallel analysis of NFATC1 in the experiment with primary cells. Even though these results would argue for a more potent action of NFATC2 during Th17, data with NFATC1 would increase this demonstration.

Page 11: The data presented in HEK293T cells indicate that NFATC2 and NF κ B cooperate in inducing RORC2 expression. Again, this in vitro assay would gain insights if the experiments were performed in a model in which TCR might be stimulated.

Figure 6D : Binding of p50 and p65 is shown upon Th17 stimulation but not upon TCR-TGF β stimulation. This would help understanding the relative contribution of NFATC1 and C2 in this differentiation process.

Minor comments

Page 8: when authors state that a weaker binding to oligos 1 and 2 was observed this is somehow overstatement since this methodology is not quantitative, sentence should be rephrased.

Page 9: It is important in the legend of supplementary Figure 2F and in the result section to specify in which cells the experiment was performed.

Reviewer #3 (Epigenetics, GWAS)(Remarks to the Author):

In this study, authors analyzed regulatory elements at the human ROR γ t locus in CD4+ T cells. The authors demonstrate that TCR stimulation calcium dependent acetylation of these elements resulting in NFAT binding and activation of ROR γ t promoter in cooperation with NF- κ B transcription factors.

The manuscript is thorough and the analysis detailed. I don't have major comments regarding the experimental design. The comments below are minor but I think they could make the paper more approachable.

Comments:

The manuscript is very lengthy and because this is a very descriptive paper it is difficult to read it. I would suggest shortening the manuscript, probably by half. I would suggest keeping only major results as main figures while other, that are more repetitive and act as controls, can be moved to the supplement. I would also suggest to shorten and focus both in introduction and discussion.

Figure 1: Can the authors provide justification for using H4 acetylation, rather than the more standard H3K27ac.

In 1B it states "mRNA levels are represented relative to levels in DN thymocytes". If everything is relative to DN thymocytes then the expression in DN thymocytes should be 1? Please provide justification for this normalisation to DN thymocytes?

Fig 2B, Fig 3 -7 most panels are very observational and lack statistics to support if observed changes are significant between the different groups.

I would suggest moving Fig 3B to supplement.

Fig 4A Why is NFATC1 binding reduced by Th17, and what is the number of biological replicates to support that?

Figure 6. What is the number of samples? Please provide error bars?

My suggestion is to either keep all the figures in colour (preferably) or black and white. Only one figure (figure 2) in colour is distracting.
Fig 2A is missing base pair coordinates.

Reply to reviewers' comments:

Reviewer #1 (T17, TF regulation, Rorgt)(Remarks to the Author):

This manuscript by Hanane et al described the function of NFAT in the regulation of expression of a functionally important transcriptional factor RORgt in Th17 cells. However, a paper by Kyun-Do Kim (Journal of Immunology, 192: 119) has described the function of Ca²⁺ channel Orai1-regulated activation of NFAT in Th17 function by regulating RORgt expression, which affects the novelty of this manuscript. In addition, this paper was not mentioned in the manuscript.

We apologize for having missed the paper by Kim and colleagues. Kim's paper focused on the role of Ca²⁺ release-activated Ca²⁺ (CRAC) channels in murine Th17 differentiation, and Fig. 4A shows NFATc2 binding to the *Rorc* promoter.

We would like to point out that the study by Kim et al. investigated NFAT activity in the mouse and strengthens our results obtained with human cells, as the authors also found that Nfatc2 binds to the *Rorc* promoter in differentiated Th17 cells. This is important not only to draw a parallel between the murine and the human system, but also in the light of the discussion of the respective roles of different NFAT proteins in RORγt induction. Interestingly, the authors show that expression of the activated form of NFATc2 (CA-NFAT) could rescue completely expression of *Rorc*, but only moderately IL-17A expression, in cells deficient for the pore component of the CRAC channel. These data support a central role for Nfatc2 in Rorc regulation, while other Ca²⁺-dependent transcription factors (possibly Nfatc1) may be important for IL-17 expression.

This publication is now cited on **page 8 and 15** of the manuscript, and is also referenced in the legend of **Supplementary Figure 4**. We could not include it in Suppl. Fig.5, since the location in the murine promoter of the NFAT binding site was not defined in the paper.

(New text in the revised manuscript and in this reply is in **red**; new figures, or figures with new experimental data are indicated in **bold**).

Page 8: **Binding of NFATc2 to the murine *Rorc* gene was described by Kim and coworkers to occur in Th17-polarizing conditions³².**

Page 15: **...consistent with this study and with the findings of Kim *et al.*³², we found that NFATc2, but not NFATc1, bound to the human *RORC* locus in differentiating Th17 cells 20h after stimulation.**

Supplementary Figure 4 legend: **NFATc2 binding to the promoter region (exact site not characterized): Kyun-Do K., Sonal S., Yossan-Var T., *et al.*2014. Calcium Signaling via Orai1 Is Essential for Induction of the Nuclear Orphan Receptor Pathway To Drive Th17 Differentiation. Journal of Immunology. 192: 110-122.**

With that said, this manuscript did reveal some very useful detailed information: 1) define epigenetic modification of RORgt locus; 2) identify some potential regulatory elements; 3) human Th17 cells and thymocytes. The problem with identified multiple regulatory elements is their

unknown physiological function. At least, is it possible to use CRISPR/CAS9 technology to mutate some sites to determine their function in the regulation of ROR γ t expression.

In summary, this manuscript provides some detailed information about NFAT function in human Th17 cells. However, due to similar conclusion by previous publication, it is likely not novel enough to be published in Nature Communications. Whereas, publication of this manuscript in a more specialized journal will benefit the related community.

We have welcomed the reviewer's suggestion, and have performed CRISPR/Cas9-mediated deletion of several elements with regulatory marks, comprised between the *RORC* and *RORC2* promoter, and tested their effect on *RORC2* transcription.

Based on our observation that NFAT binding sites seem to tag clusters of transcription factors (TF) binding sites that are important for Th17 differentiation (see Supplementary Fig.5 and discussion), we decided to delete these clusters, at the putative enhancer region we had identified in our epigenetic studies, and the *RORC2* promoter. We deleted CNS -2.5 and CNS -1.8 individually (**Fig. 7C** and **Supplementary Fig 5**, O1 and O2), and both in combination (**Supplementary Fig 5**, O1/2). We also tested the functional importance of the region containing the O4 and O5 NFAT binding sites, which is poorly conserved, and was not yet characterized in the literature. As a positive control we introduced a deletion of the TF sites cluster in the proximal *RORC2* promoter (O6/7), and a larger one of the extended promoter region (O4/7).

Successful deletion of these regions was confirmed by PCR amplification of genomic fragments containing the targeted site, an approach also applicable to the estimation of deletion frequencies in bulk cultures of CRISPR/Cas9-transfected primary cells.

We first carried out the CRISPR/Cas9 experiments in the Jurkat clone E6.1. The advantage of this approach is the possibility of obtaining stable, well-characterized clones carrying the genomic deletion. However, we were conscious of the fact that *RORC2* regulation in Jurkat cells may not reflect the regulation of *RORC2* expression in normal peripheral CD4⁺ T cells. We found that *RORC2* is constitutively expressed in Jurkat cells, at levels that depend on the cell concentration, and was not increased by TCR stimulation. This behaviour may be more closely related to *RORC2* regulation in thymocytes, consistently with the origin of this leukemic cell line.

The experiments in Jurkat cells allowed us to screen crisprRNAs for their deleting efficacy in transfected bulk cultures, and to confirm the cutting sites by DNA sequencing, after amplification of the selected target sites. However, single cell cloning revealed a large clonal variability in *RORC2* expression in wild type and Cas9 control Jurkat cells. This allowed us to perform functional assays only on selected deletions (an example is shown in **Supplementary Fig. 6A**), for which we could obtain a sufficient number of clones for a statistical analysis.

We therefore decided to address the question in primary human CD4⁺ T cells, using a different approach adapted to the transfection of primary human lymphocytes. Freshly isolated CD4⁺ T cells were transfected with CRISPR/Cas9 riboprotein complexes based on the crisprRNAs validated in Jurkat cells (see **Fig. 7D** and **Supplementary Fig. 6B and 6C**). We

tested the effect of the deletions by measuring directly the number of transcripts for RORC2 and for an additional panel of immune-related genes using nCounter technology (NanoString), which avoids the need for enzymatic manipulation of RNA (reverse transcription and amplification) and, in our hands, shows high sensitivity coupled with very high reproducibility.

The results showed that deletion of O1 or O2 independently affects RORC2 expression, and that deletion of both sequences together has an additive effect on RORC2 transcript levels. Deletion of O4/5 also reduced RORC2 expression, indicating that also this region of the promoter plays a role in human RORC2 regulation, despite its overall limited conservation in the mouse (**Fig.7 C-E**).

These results support a key role of the genomic regions analysed in regulating RORC2 expression.

An additional interesting outcome of these genomic deletions and the resulting RORC2 suppression, was an increase in a number of Th1-related transcripts, including *IFNG*, *GZMB*, *CCR5*, *TNFSF8*. This suggests that blocking Th17 differentiation unmasks additional effects of TCR stimulation and of the Th17-inducing cytokines, in particular IL-23, which was shown to also activate STAT4 (Oppmann et al. *Immunity*, 2000). IL-23 was also shown to expand a memory, CD45Rb^{low} T cell population in the mouse, which may explain the decrease in *CD45RB* expression observed in populations with genomic deletions of RORC2. These findings support a close relatedness of Th1 and Th17 cells, and are in agreement with the plasticity displayed by Th17 cells towards the Th1 subset upon repeated stimulation.

Described on Page 11-12:

Deletion of CNS-2.5 and CNS-1.8 result in reduced RORC2 transcription.

The analysis of histone modifications and the transcriptional regulators binding data suggest that the genomic regions corresponding to CNS-2.5 and CNS-1.8 may have an enhancer function on *RORC2* expression. These data, together with data from the literature (Supplementary Fig. 4 and legend), suggest that these regions are important to nucleate the assembly of clusters of transcription factors that integrate stimuli from the TCR and from cytokines to regulate *RORC2* expression. To explore the functional significance of these regions in promoting *RORC2* transcription, we used CRISPR/Cas9 technology to engineer genomic deletions of *RORC2* regulatory regions in human CD4⁺ T lymphocytes and study their effect on *RORC2* transcription. We deleted CNS-2.5 and CNS-1.8 individually (O1 and O2), and both in combination (O1/2), and also tested the functional importance of the region containing the O4 and O5 NFAT binding sites (Fig. 7C and Supplementary Fig. 5A and B). As a positive control we introduced a deletion of the proximal *RORC2* promoter (O6/7), and a larger one of the extended promoter region (O4/7, Fig. 7C and Supplementary Fig. 5A). The choice of guide RNAs was based on high target specificity and low number of off-target sites (Supplementary Table 1F). Targeting efficacy was tested in Jurkat cells (Supplementary Fig.6A).

Transfection of crRNA/Cas9 riboprotein complexes in human primary CD4⁺ T cells achieved deletion of all the targeted regions at variable percentages in 4 different donors, as confirmed by PCR amplification of the genomic target region (Fig. 7D and Supplementary Fig. 6B-D). We used nCounter technology (Nanostring) to quantify the effects of the genomic deletions on

the expression of *RORC2* and of a panel of immune-related genes. Analysis of *RORC2* expression showed significantly reduced absolute counts of *RORC2* transcripts in cells carrying deletions of the *RORC* regulatory regions, both at the promoter and at the enhancer, compared to cells transfected with the negative control ribocomplex (Fig. 7E). Contemporary deletion of CNS-2.5 and CNS-1.8 resulted in increased *RORC2* inhibition, arguing for an additive function of these regions. Deletion of the region containing NFAT sites O4/5 also resulted in decreased levels of *RORC2* transcripts, suggesting that this poorly conserved region may also play a role in human *RORC2* regulation (Fig. 7C and E).

Multigroup comparison of gene expression in the control and samples carrying genomic deletions showed that a number of other immune genes were affected by the genomic deletions and reduced *RORC2* expression (Fig. 7F, Supplementary Fig. 7). In particular, we noticed an elevated IFN γ expression in CRISPR/Cas9-deleted samples, possibly due to stimulation by IL-23 present in the Th17-inducing cytokine cocktail and activation of STAT4³⁵. Also increased in samples with genomic deletions at the *RORC* locus were transcripts from a number of IFN γ target genes, or genes associated with IFN γ -producing cells (*PTPN22*, *TNFSF8/CD153*, *CDKN1A/p21Cip*, *CCR5*, *GZMB*)³⁶⁻³⁸, suggesting a skewing towards a Th1-type response in cells stimulated in Th17-inducing conditions when ROR γ t expression is inhibited. *PTPN22* was also described as a direct target of ROR γ t³⁹. These findings support an enhancer role for the CNS-2.5 and CNS-1.8 sequences.

Reviewer #2 (NFAT, signalling)(Remarks to the Author):

The manuscript entitled “NFAT primes the human RORC locus for ROR γ t expression in CD4+ T cells” by Yahia-Cherbal et al. investigates the factors involved in the transcriptional regulation of Th17 lineage. Using a number of epigenetic and transcriptomic approaches they explore the complex multifactorial interplay of transcription factors and chromatin conformational changes involved in ROR γ t expression and the subsequent differentiation of Th17 cells. Analysis covers transcriptional and conformational changes in thymic and peripheral cells.

The article investigates a rather challenging question with an elegant demonstration of the key role played by the NFAT family of transcription factors in both transcriptional regulation and chromatin permissive conformation. An important number of experiments show specific roles for NFATc2 in both the direct transcriptional control of ROR γ t and the conformational opening necessary to CD4+ T cells differentiation.

Importantly, experiments are mainly performed using primary cells from thymus and cord blood. A detailed analysis of the RORC and RORC2 promoter regions is presented and gives important clues on the specific or more general mechanisms involved in the regulation.

Moreover the results give important insights on the differential impact of TCR- dependent or Th17 specific cytokines-dependent mechanisms in the activation of ROR γ t expression. The manuscript is sound; results are clear and well discussed. Important specific Th17 differentiation process is analysed in the periphery. The main and almost single concern of the reviewer is the relative impact of the different members of the NFAT family in this regulation. Despite a number of arguments well discussed in the discussion of the manuscript, the relative impact of TCR-NFATc1 versus Th17-NFATc2 mediated process would need further assessment in several experiments of NFATc1 binding and transcription mediated regulation in primary cells. The question is challenging due to a number of reciprocal regulations between NFATc1 and NFATc2, which is very well discussed.

However the time-related effect of both factors should be addressed regarding the literature on these processes.

We have studied the kinetics of NFATc1 versus NFATc2 activity in the initial phase of Th17 differentiation of cord blood derived CD4⁺ T cells.

Using Imaging Flow Cytometry (ImageStream), we quantified nuclear translocation of NFATc1 and NFATc2 proteins in cord blood CD4⁺ T cells from 4 donors (**Figure 4A-D, Supplementary Fig. 2**).

Fig. 4C shows that early after stimulation, NFATc2 underwent significant nuclear accumulation, which was then maintained in a lower percentage of cells at 20 and 40 hours after stimulation. NFATc2 translocation preceded temporally the appearance of *RORC2* transcripts (Supplementary Fig. 1A). Nuclear translocation of NFATc1, instead, followed a slower kinetics. No differences in NFAT nuclear accumulation were observed between samples stimulated in Th17 or the TCR+TGFβ conditions, indicating that TCR stimulation is likely the main driver of nuclear translocation.

We have used NanoString technology to quantify *NFATC1* and *NFATC2* mRNA expression at different time of stimulation in naïve CD4⁺ T cells from 4 different cord blood (**Figure 4E**). The advantage of this approach is the possibility to obtain absolute mRNA counts, rather than a relative quantification, allowing a direct comparison between *NFATC1* and *NFATC2* mRNA levels. *NFATC2* mRNA levels were higher in naïve cells than *NFATC1* mRNA, as previously reported, and remained higher for the duration of the observation. The early changes in mRNA levels were not immediately translated in changes in protein levels, suggesting an additional layer of control in NFAT proteins expression. ImageStream analysis showed that both Th17 and TCR+TGFβ stimulation resulted in a significant increase of NFATc2 protein (**Fig. 4D**), which persisted for the time of observation, while the increase of NFATc1 mean fluorescence intensity, although statistically significant, was smaller.

Described on Page 8: To explore the role of the NFAT pathway in human Th17 differentiation, we studied the early dynamics of NFAT nuclear translocation in differentiating cord blood CD4⁺ T cells. Imaging Flow Cytometry analysis demonstrated a strong nuclear accumulation of NFATc2 in cells stimulated in Th17-differentiating conditions, as well as in cells stimulated through the TCR in the presence of TGFβ (Figure 4B, 4C, Supplementary Fig. 2), at an early time point, preceding the increase in *RORC2* mRNA levels (Supplementary Fig. 1A). Nuclear translocation of NFATc1 followed a slower kinetics, and was less pronounced than NFATc2 translocation. Both Th17 and TCR+TGFβ stimulation resulted in an increase of *NFATc2* mRNA and protein (Fig. 4D, Supplementary Fig. 1C, 1D), which persisted for the time of observation, while the transient increase of *NFATc1* mRNA (Fig. 4E) was accompanied by a smaller, although significant increase in NFATc1 mean fluorescence intensity.

Reviewer requests:

In Figure 4, an important TCR- TGFβ impact is shown on NFATC1 binding to RORC2 promoter while NFATC1 is not expressed at the highest levels. Th17 cytokines show more impact on NFATC2

binding. Therefore, results shown in Figure 4F would be strengthened by the parallel analysis of NFATC1 in the experiment with primary cells. Even though these results would argue for a more potent action of NFATC2 during Th17, data with NFATC1 would increase this demonstration.

(This comment refers to **Fig. 5** of the revised manuscript)

We have performed the required experiment, and it is shown in **Supplementary Fig. 3E**.

We find that, just like NFATc2, NFATc1 present in lysates from primary CD4⁺ T cells can bind *in vitro* to oligos containing the *RORC2* NFAT sites, regardless of the type of stimulation the cells had received before lysis. This is perhaps not surprising, since NFATc1 and NFATc2 are known to be able to recognize the same consensus sequence (as also shown by our DNA affinity capture experiment with Jurkat lysates, Suppl. Fig.3C and Fig. 5C), and in the *in vitro* binding conditions the amount of free NFAT binding sites is not limiting, reducing a possible competition between NFAT factors for binding to the same site.

As this reviewer correctly pointed out, this is not a quantitative assay, but it is useful to confirm that the *in silico* predicted sites are genuine binding sites for NFAT.

Described on Page 9: A similar pattern of binding to the *RORC* oligos was detected when protein lysates were obtained from primary CD4⁺ T cells isolated from cord blood (Fig. 5E, **Supplementary Fig. 3E**).

Page 11: The data presented in HEK293T cells indicate that NFATC2 and NFKB cooperate in inducing RORC2 expression. Again, this *in vitro* assay would gain insights if the experiments were performed in a model in which TCR might be stimulated

We chose the HEK293T model because of the possibility of testing the effect of expressing NFATc1 and NFATc2, without the confounding contributions from endogenous NFAT proteins.

We have also attempted some reporter gene experiments in primary human CD4⁺ T cells to test the activity of the enhancer regions, and the effect of stimulation on this activity. However, the transfection efficiency of the reporter plasmids was low and showed great variability, and cell mortality elevated, so that the results were not conclusive.

The following figure shows a summary of these results, obtained by transfecting (Amaxa) reporter constructs containing either the *RORC2* promoter, or the promoter together with different CNS sequences located in the enhancer region upstream of the *RORC2* promoter (average and SD of 3-4 experiments for each construct).

The differences in luciferase activity between the different constructs or stimulation conditions were not statistically significant and the results were not included in the manuscript.

Figure 6D: Binding of p50 and p65 is shown upon Th17 stimulation but not upon TCR-TGF β stimulation. This would help understanding the relative contribution of NFATC1 and C2 in this differentiation process.

(This comment refers to **Fig. 8** of the revised version)

In the first version of the manuscript we used ChIP assay to show binding of NF-kB p50 to the *RORC2* locus in cells stimulated in Th17 conditions. Following the reviewer's suggestion, we have now performed ChIP assays to compare binding of NF-kB to *RORC2* in cord blood-derived CD4⁺ cells either in Th17-inducing conditions or upon TCR-TGF β stimulation.

To perform these assays, we did not have sufficient amounts of the original anti-p50 Santa Cruz polyclonal antibody used for the previous assay, as polyclonal antibodies are no longer available from this manufacturer. In addition, none of the anti-p50 antibodies tested (sc-8414X, sc114x, Millipore 06-866-1) gave reliable results in our ChIP assays.

We were happy, however to find that of the 7 antibodies tested for the p65 NF-kB subunit, a Millipore monoclonal (MAB3026) gave clear, reproducible results, shown in **Fig. 8B** of the revised manuscript. We could demonstrate in three independent experiments that p65 binds to the *RORC* locus exclusively in cells stimulated in Th17-inducing conditions, and no binding of p65 was detected in TCR-stimulated cells in the presence of TGF β alone.

These data clearly demonstrate that the transcriptional complex assembled at the *RORC* locus is different in the two conditions. Taken together with the reporter gene assays data, the findings support the notion that full transcriptional competence is achieved when both NFATc2 and p65 are present at the *RORC2* promoter. The presence of p65 and possibly additional transcriptional regulators may stabilize NFATc2 binding at the locus, in competition with NFATc1 binding. Although of interest, the identification of all the components of this complex will require a different proteomic approach, and goes beyond the scope of this manuscript.

Described on Page 12-13: ChIP experiments in CD4⁺ cord blood T cells with an antibody against the p65 NF-kB subunit confirmed binding of NF-kB to the *RORC2* promoter, **as well as to upstream enhancer (CNS-1.8 Fig. 8B)**. Notably, **p65 binding was detectable in cells stimulated in Th17 conditions, but not in naïve unstimulated cells, nor in TCR-stimulated cells in the presence of TGF β alone**. These data indicate that different transcriptional complexes assemble at the *RORC2* promoter in different stimulation conditions (TCR+TGF β versus Th17) and that Th-inducing cytokines are important for NF-kB binding to the *RORC2* locus and full activation of human *RORC2* expression.

In conclusion, we believe that our data are in agreement with a preferential role for NFATc2 in the early steps of Th17 differentiation: direct evidence is provided by preferential

binding of NFATc2 in ChIP assays in Th17-inducing conditions, and NFATc2-mediated transcriptional activation in reporter gene assays. Indirect evidence are the kinetics of NFATc2 nuclear translocation and of NFATc2 protein accumulation in naïve CD4⁺ T cells, which are consistent with a more pronounced activity of NFATc2 versus NFATc1 in the early activation of these cells.

These findings are further supported by data in the literature generated in mouse models. In addition to the binding of NFATc2 to the *Rorc* promoter described by Kim *et al.* (see above), Ghosh *et al.* (PNAS, 2010) describe a similar early role for constitutively activated NFATc2 (CA-NFAT) in the early *Rorc* induction (peak at 8hr).

As discussed in our manuscript, the interpretation of NFAT knock-out experiments is more complicated, due to the reciprocal regulation of NFATc1 and NFATc2, which shows a compensatory activity of one protein when the other is deleted. This is the very plausible explanation for the increased *Rorc* expression observed by Dietz *et al.* (EJI, 2015) in the mouse *Nfatc2* ko model, which also presents a marked *Nfatc1* nuclear accumulation. *Nfatc1* deletion in this work resulted only in a small, although statistically significant, decrease in *Rorc* transcripts (Fig. 3D of Dietz *et al.*), however it clearly impaired Th17 cell differentiation. This may be due to other roles of NFATc1 in this pathway, in particular in the induction of IL-17 by direct binding to the *Il17* locus (Gomez-Rodriguez, Immunity, 2009; Purvis *et al.* Blood, 2010; Liu X.K. *et al.* JBC, 2004). Our manuscript focusses on the early events of Th17 differentiation, but NFATc1 is likely to be important for further expansion and survival of differentiated Th17 cells, given its ability to directly control the *Il2* locus and many genes involved in crucial metabolic pathways, as accurately described for CD8⁺ T cells by Klein-Hessling *et al.* (Nature Comm, 2017).

Minor comments

Page 8: when authors state that a weaker binding to oligos 1 and 2 was observed this is somehow overstatement since this methodology is not quantitative, sentence should be rephrased.

The phrase was changed to: “**We also observed binding to oligos 1 and 2**” (Page 9 of the revised manuscript).

Page 9: It is important in the legend of supplementary Figure 2F and in the result section to specify in which cells the experiment was performed.

The experiments were performed in HEK293T cells. This has been reiterated in the text (page 9 and 10), and mentioned in the legend of Supplementary Fig. 3.

Reviewer #3 (Epigenetics, GWAS)(Remarks to the Author):

In this study, authors analyzed regulatory elements at the human ROR γ t locus in CD4⁺ T cells. The authors demonstrate that TCR stimulation calcium dependent acetylation of these elements resulting in NFAT binding and activation of ROR γ t promoter in cooperation with NF-kB transcription factors.

The manuscript is thorough and the analysis detailed. I don't have major comments regarding the

experimental design. The comments below are minor but I think they could make the paper more approachable.

Comments:

The manuscript is very lengthy and because this is a very descriptive paper it is difficult to read it. I would suggest shortening the manuscript, probably by half. I would suggest keeping only major results as main figures while other, that are more repetitive and act as controls, can be moved to the supplement. I would also suggest to shorten and focus both in introduction and discussion.

We agree with the reviewer that the discussion of this manuscript is perhaps longer than usual. However, we think it has the value to provide a coordinated view of the regulatory hubs at the RORC2 locus.

For the resubmission of the paper we have tried to streamline the wording where possible, but we have followed the comment from the editor that significantly shortening the manuscript was not necessary. We will be happy, however, to shorten the manuscript if required.

Figure 1: Can the authors provide justification for using H4 acetylation, rather than the more standard H3K27ac.

In this study we have used both H4 acetylation, as a mark associated with actively transcribed regions in T lymphocytes (ex. P. E. Fields, Sean T. Kim and Richard A. Flavell, J Immunol July 15, 2002) and have also analysed H3K27 acetylation as a marker of active enhancers (Zhou et al. Nat. Reviews Genetics, 2011)

In 1B it states “mRNA levels are represented relative to levels in DN thymocytes”. If everything is relative to DN thymocytes then the expression in DN thymocytes should be 1? Please provide justification for this normalisation to DN thymocytes?

The scale has now been set to 1 (instead of 100 as it was in the previous figure). This Real-Time PCR experiment measures relative levels of transcript, and requires that one sample should be used as reference. For ease of reading, we often choose the sample with the highest Ct as reference, and represent the other values as fold-increases. In this case, this fits nicely with the fact that DN thymocytes are the most undifferentiated thymic population, so that the samples are ordered by developmental age.

Fig 2B, Fig 3 -7 most panels are very observational and lack statistics to support if observed changes are significant between the different groups

We agree that the analysis of epigenetic marks is mainly descriptive. We have used it as tool to pinpoint genomic regions at the RORC2 locus for further functional characterization. This has been performed by identifying in these regions possible binding sites for transcription factors, and verifying their binding. In the revised version of the manuscript we further characterize the role of the putative regulatory regions, by the introduction of CRISPR/Cas9-engineered deletions (see response to reviewer 1)

I would suggest moving Fig 3B to supplement.

Fig. 3 consists of only one panel. We feel that it carries important information, because it shows that CNS -2.5 and CNS -1.8 are enriched in histone modification associated with active enhancers, and it is the first indication that they may represent regulatory regions.

Fig 4A Why is NFATC1 binding reduced by Th17, and what is the number of biological replicates to support that?

(We think that this comment probably referred to Fig 4B, which is **Fig. 5A** of the revised version)

The model we propose is that the transcription factors complex that forms at the RORC2 locus in the presence of the Th17-inducing cytokines stabilizes binding of NFATc2, rather than NFATc1 (see comments to referee #2). Although the whole complex requires further characterization, one component that is uniquely present at the RORC2 promoter in Th17-inducing conditions (and absent in TCR+TGF β conditions) is the NF-kB factor p65 (see **Fig. 8B**)

Fig. 4B (now **Fig. 5A**) shows the average of technical replicates of one representative experiment of three (SD too small to be visible). A second experiment is shown in Supplementary Fig. 3A.

Figure 6. What is the number of samples? Please provide error bars?

(Now **Fig. 7A**, B and **Fig. 8B**)

The number of samples is now described in the Figure legends. Fig. 7A shows two representative experiments of four for p300, and two of three for CBP (the other experiments are provided in **Supplementary Fig. 3J** and **K**). The ChIP experiments were performed with 2 different antibodies or different lots of the same antibody, and although the binding pattern is conserved the percentage of input varies in the different experiment. We therefore show them as separate experiments.

The Figure for the NF-kb ChIP originally in panel 6D has been now substituted by **Fig. 8B**, with expanded stimulation conditions, as requested by reviewer 2. This panel shows average and SD of 3 independent samples.

My suggestion is to either keep all the figures in colour (preferably) or black and white. Only one figure (figure 2) in colour is distracting.

We now provide figures with similar content (**Fig. 1** and **Fig. 3**) in colour.

Fig 2A is missing base pair coordinates.

Chromosomal coordinates were added to Figure 2A.

REVIEWERS' COMMENTS:

Reviewer #1 (Remarks to the Author):

With the functional analysis of two major NFAT-binding sites in the regulation of IL-17A expression, this manuscript is greatly improved. Although the function of NFAT in Th17 has been shown in the differentiation of Th17 in mice, which affects the novelty of this manuscript, this study still provides useful information in addition to confirmation of previous studies.

Reviewer #2 (Remarks to the Author):

The authors performed a number of additional experiments as requested or suggested by the reviewers. The requests raised by reviewer 2 were carefully addressed and substantially strengthened the original statements and conclusions.

Several technical constraints (lack of antibodies) prevented the authors from investigating one concern. Therefore the revised manuscript might be considered for publication at this stage.

One minor revision should nevertheless be mentioned: since the manuscript is rather extended the authors might consider revising and clarifying the figure legends with a specific shortening of the new figures legends

Reviewer #3 (Remarks to the Author):

I'm satisfied with the revised manuscript.

REVIEWERS' COMMENTS:

Reviewer #1 (Remarks to the Author):

With the functional analysis of two major NFAT-binding sites in the regulation of IL-17A expression, this manuscript is greatly improved. Although the function of NFAT in Th17 has been shown in the differentiation of Th17 in mice, which affects the novelty of this manuscript, this study still provides useful information in addition to confirmation of previous studies.

Reviewer #2 (Remarks to the Author):

The authors performed a number of additional experiments as requested or suggested by the reviewers. The requests raised by reviewer 2 were carefully addressed and substantially strengthened the original statements and conclusions. Several technical constraints (lack of antibodies) prevented the authors from investing one concern. Therefore the revised manuscript might be considered for publication at this stage.

One minor revision should nevertheless be mentioned: since the manuscript is rather extended the authors might consider revising and clarifying the figure legends with a specific shortening of the new figures legends

Figure legends have been shortened

Reviewer #3 (Remarks to the Author):

I'm satisfied with the revised manuscript.